# Efficient all-small-molecule organic solar cells processed with non-halogen solvent

Wei Gao[1,9], Ruijie Ma [2,9] ✉, Top Archie Dela Peña [3,4,9], Cenqi Yan[5], Hongxiang Li [5] ✉, Mingjie Li [3], Jiaying Wu [4], Pei Cheng [5], Cheng Zhong[6], Zhanhua Wei [1], Alex K.-Y. Jen [7,8] ✉ & Gang Li [2] ✉

All-small-molecule organic solar cells with good batch-to-batch reproducibility combined with non-halogen solvent processing show great potential for commercialization. However, non-halogen solvent processing of all-small-molecule organic solar cells are rarely reported and its power conversion efficiencies are very difficult to improve. Herein, we designed and synthesized a small molecule donor BM-ClEH that can take advantage of strong aggregation property induced by intramolecular chlorine-sulfur non-covalent interaction to improve molecular pre-aggregation in tetrahydrofuran and corresponding micromorphology after film formation. Tetrahydrofuran-fabricated all-small-molecule organic solar cells based on BM-ClEH:BO-4Cl achieved high power conversion efficiencies of 15.0% in binary device and 16.1% in ternary device under thermal annealing treatment. In contrast, weakly aggregated BM-HEH without chlorine-sulfur non-covalent bond is almost inefficient under same processing conditions due to poor pre-aggregation induced disordered π-π stacking, indistinct phase separation and exciton dissociation. This work promotes the development of non-halogen solvent processing of all-small-molecule organic solar cells and provides further guidance.

Organic solar cells (OSCs), featuring unique merits such as lightweight, mechanically flexible and semitransparent, have shown great promising in future smart city[1-4]. Due to continuous innovations in photoactive materials[5-12], device optimizations[13,14] and interface regulations[15,16], OSCs have been witnessed going through rapid progresses and achieved over 19% power conversion efficiency (PCE) recently[17-26], which significantly narrows the PCE gaps with other

photovoltaic technologies[27,28]. Compared to OSCs based on polymer donor (PD):small molecule acceptor (SMA), OSCs enabled by all-small-molecule (ASM) materials is generally considered to be more promising in terms of practical applications because small molecule donor (SMD) component is provided with well-defined chemical structures and thus can well overcome the batch-to-batch difference of PDs in molecular weight changes caused by difficult control of

[1]Xiamen Key Laboratory of Optoelectronic Materials and Advanced Manufacturing, Institute of Luminescent Materials and Information Displays, College of Materials Science and Engineering, Huaqiao University, Xiamen 361021, China. [2]Department of Electrical and Electronic Engineering, Research Institute for Smart Energy (RISE), The Hong Kong Polytechnic University, Hung Hom, Kowloon, Hong Kong 999077, China. [3]Department of Applied Physics, The Hong Kong Polytechnic University, Hong Kong 999077, China. [4]Advanced Materials Thrust, Function Hub, The Hong Kong University of Science and Technology, Nansha, Guangzhou 511442, China. [5]College of Polymer Science and Engineering, State Key Laboratory of Polymer Materials Engineering, Sichuan University, Chengdu 610064, China. [6]Department of Chemistry, Hubei Key Lab on Organic and Polymeric Optoelectronic Materials, Wuhan University, Wuhan 430072, China. [7]Department of Materials Science and Engineering, City University of Hong Kong, Kowloon, 999077 Hong Kong, China. [8]Hong Kong Institute for Clean Energy, City University of Hong Kong, Kowloon, 999077 Hong Kong, China. [9]These authors contributed equally: Wei Gao, Ruijie Ma, Top Archie Dela Peña. ✉e-mail: ruijie.ma@polyu.edu.hk; lihongxiang@scu.edu.cn; alexjen@cityu.edu.hk; gang.w.li@polyu.edu.hk

polymerization reactions[29]. Benefiting from satisfactory properties of Y6 and its derivatives as well as unremitting efforts in molecular structures improvement of SMDs, ASM-OSCs nowadays are receiving increasing attentions and show great advances in PCEs[30-37].

As the PCE limitation is gradually eliminated, the focus of OSCs is developing towards standards suitable for commercial applications. Among them, replacing halogen solvents in the processing of OSCs is a required step. Over the years, considerable efforts have been put into exploring non-halogen solvent processing for OSCs including material design, film forming kinetics, micromorphology optimization and efficiency improvement[38,39]. Recent advances show that non-halogen solvent processed OSCs enabled by PD:SMA, all-polymer and all-small-molecule systems have achieved over 18%, 17% and 14% PCEs, respectively[40-42]. Even so, it is not difficult to find that currently state-of-the-art OSCs, whether based on PDs or SMDs, rely heavily on halogen solvents, especially chloroform (CF)[17-26,30-37], because of its excellent solubility for a wide variety of photovoltaic materials and proper volatility for favorable kinetics of film formation. However, CF possesses obvious disadvantages of unavoidable biological toxicity and high energy-consuming synthesis, which makes it unfavorable in terms of sustainable development and will severely limit the large-scale production of OSCs. In view of this, a lot of attempts have been made to search suitable non-halogen solvents instead of halogen solvents for processing active layers of OSCs. Many good results with competitive PCEs relative to ones processed by halogen solvent have been achieved when employing toluene or o-xylene as non-halogen processing solvent[43]. However, it seems that non-halogen solvent processing method is found to be successful only in PDs-based OSCs rather than in SMDs-based OSCs[38-45]. Although ASM-OSCs enabled by benzodithiophene trithiophene radanine (BTR)-series and Y6 derivatives are able to realize over 17% PCE[33,34,36,37], their non-halogen solvent-processed devices have been rarely reported[42].

Unlike high-molecular-weight PDs that exhibit stronger intermolecular π-π stacking and entanglement induced by long conjugated backbones, SMDs typically have shorter main chains with limited π-conjugations, which will cause ASM-OSCs being more sensitive to changes of processing solvent[44]. The obviously property differences in molecular size, polarity and volatility between non-halogen solvent and halogen solvent are very likely to have negative effects on pre-aggregation state in solution and film-forming kinetics of SMDs, which may be the reason why ASM-OSCs fabricated by non-halogen solvents is difficult to success. The most feasible way is to enhance intermolecular interaction of SMDs so that they are not easily diluted by solvent molecule and try to maintain good pre-aggregation state in solution[45-47]. In fact, the interaction between SMDs completely depends on self-aggregation of molecules, which is quite different from entanglement effect between PDs. In this sense, designing and synthesizing SMDs with strong aggregation property will have great potential for realization of high-performance non-halogen solvent-processed ASM-OSCs.

In our previous work, Cl atom was rationally introduced into the molecular backbone of a BTR derivative, namely BM-Cl, to construct a strong intramolecular Cl-S non-covalent interaction, enabling SMD to exhibit a strong aggregation property[48]. However, sudden increase of molecular aggregation will significantly reduce the solubility of BM-Cl in non-halogen solvent where tetrahydrofuran (THF) is chosen considering its saturated vapor pressure that relates to volatility is similar to that of CF for a proper film-forming kinetics. With this in mind, we intended to extend the alkyl chain length on rhodamine (Rh) group of BM-Cl from n-butyl to 2-ethylhexyl and synthesized BM-ClEH in order to improve its solubility in THF without excessively sacrificing aggregation nature. SMD BM-HEH without Cl-substitution was also synthesized as a control of weak aggregation to better elucidate the importance of strong aggregation property of SMD for the success of non-halogen solvent-processed ASM-OSCs. BM-ClEH exhibits good solubility in THF and maintains a strong aggregation property in both CF and THF solvents since alkyl chain change on terminal Rh group does not have much influence on the rigidity of molecular skeleton. Thus, BM-ClEH forms stable pre-aggregation state in THF and strong intermolecular interaction after film formation, helping achieve a favorable micromorphology that promotes exciton dissociation, charge transport and reduces charge recombination of active layer. As a result, THF-processed ASM-OSCs based on BM-ClEH:BO-4Cl achieved a very high PCE of up to 15.0% under thermal annealing (TA) post-treatment. Furthermore, by doping a small amount (10% weight ratio) of B1[49] into BM-ClEH:BO-4Cl, THF-processed ternary ASM-OSCs delivered a PCE as high as 16.1%. In contrast, ASM-OSCs based on BM-HEH:BO-4Cl processed by THF gave no more than 1% PCE attributed to indistinct phase separation and exciton generation. Such obvious PCE difference strongly suggests that strong aggregation property of SMDs is very critical for realizing efficient ASM-OSCs fabricated by non-halogen solvent. This work aims to promote the progress of ASM-OSCs based on non-halogen solvent processing and provide some guides for further development.

## Results

### Molecular properties

Solubility test experiment showed that BM-Cl dissolves poorly in THF, which severely limits its further application into non-halogen solvent processed ASM-OSCs. In order to take advantage of its strong aggregation property, we decided to alter the alkyl chain on Rh of BM-Cl from n-butyl to 2-ethylhexyl considering such modification may have minimal impact on molecular properties. SMD BM-ClEH whose structure is depicted in Fig. 1a was designed and efficiently synthetized via Knoevenagel condensation reaction between compound 1 and 3-(2-ethylhexyl)-2-thioxothiazolidin-4-one according to synthetic route shown in Supplementary Fig. 1. Meanwhile, BM-HEH without Cl atom substitution was also prepared as a control molecule. Solubility of two SMDs (BM-ClEH and BM-HEH) in THF were estimated by gradually adding solvent to dissolve a compound quantitatively, which are both estimated to exceed 30 mg/mL (0.15 mL THF can completely dissolve 5 mg compound), well meeting the solubility requirement for processing active layer in THF (generally 10 mg/mL)[50,51].

Density functional theory (DFT) calculations were performed to understand effects of Cl-substitution on SMDs on a microcosmic aspect where long alkyl chains were shortened to be ethyl (Supplementary Fig. 2). In optimal molecular configuration, the space distance of Cl and S atoms on adjacent two thiophenes within BM-ClEH (BM-Cl) is found to be 304 pm, significantly smaller than the sum (361 pm) of van der Waals radii of S (185 pm) and Cl (176 pm) atoms, indicating the existence of intramolecular Cl-S non-covalent interaction[52]. Moreover, by further mapping intramolecular weak interactions within BM-ClEH (BM-Cl), we observed that Cl-S interaction is remarkably larger than that of H-S, revealing Cl-substitution is favorable to endow stronger intramolecular non-covalent interaction. With this effect, BM-ClEH obtains a slightly flatter molecular skeleton and a smaller binding energy of bimolecular than these of BM-HEH (−5.53 eV for BM-ClEH and −5.15 eV for BM-HEH), which suggests that Cl-substitution is conducive to realize better intermolecular aggregation/packing.

Differential scanning calorimetry (DSC) measurements were performed to evaluate changes in crystallization properties after alkyl chain extension. Corresponding DSC curves are shown in Fig. 1b. It was clearly observed that the melting point ($T_m$) dropped from 253 °C to 169 °C along with melting enthalpy ($\Delta H_m$) decreasing from 73.7 to 45.2 J/g as the alkyl chain become from n-butyl to 2-ethylhexyl, indicative of a proper crystallization performance reducing from BM-Cl to BM-ClEH. However, BM-HEH without Cl-substitution exhibits two broad melting peaks located at 155 and 135° C with lower $\Delta H_m$ of 20.7 and 24.7 J/g, respectively, which reflects a weaker intermolecular interaction and more disordered molecular packing of BM-HEH

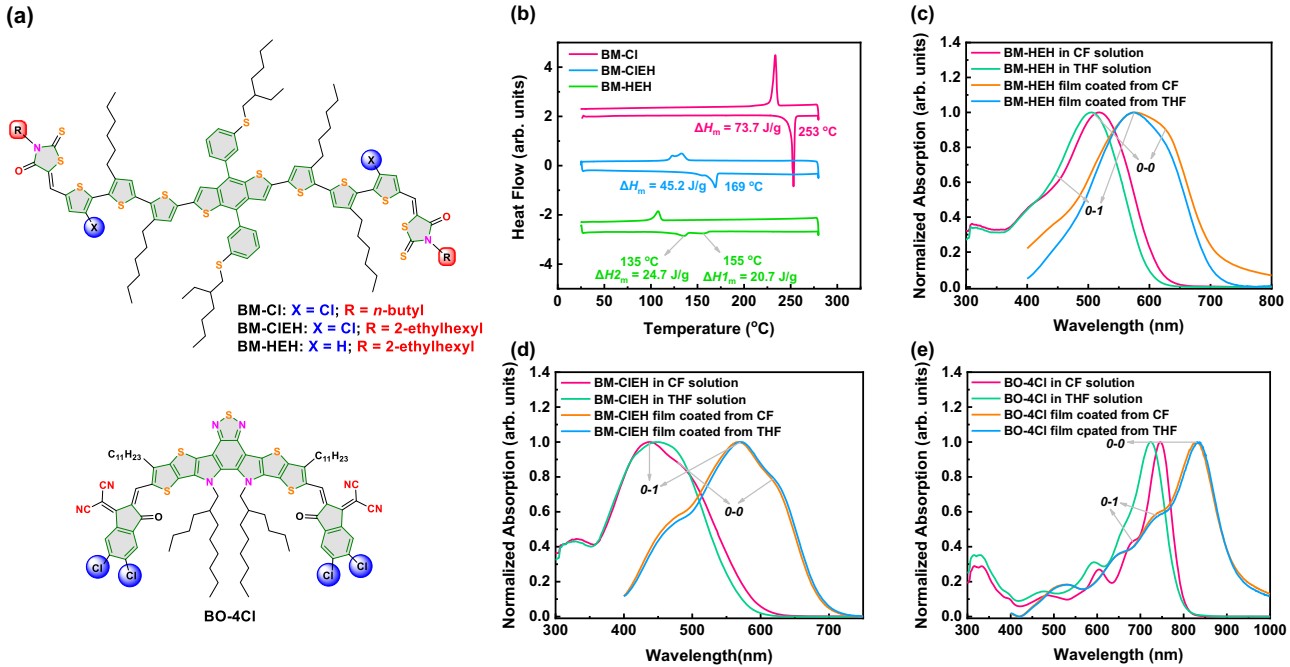

**Fig. 1 | Chemical structures, crystalline property and UV-vis absorption spectra of studied materials. a** Chemical structures of BM-Cl, BM-ClEH, BM-HEH and BO-4Cl. **b** DSC curves of BM-Cl, BM-ClEH and BM-HEH. UV-vis absorption spectra of solution and films for (**c**) BM-HEH, (**d**) BM-ClEH and (**e**) BO-4Cl.

compared to BM-ClEH. Moreover, we performed a consecutive three-lap DSC scans of three SMDs (Supplementary Fig. 3) to find that BM-Cl always maintains a sharp melting peak, whereas BM-ClEH appears another dominant melting peak and BM-HEH shows three to four flattened melting peaks. These results imply that 2-ethylhexyl on end group is easy to make SMD turn into amorphous, while intermolecular Cl-S interaction can suppress this effect to a certain extent. Thereby, BM-ClEH can achieve a compromise between ordered molecular assembly/aggregation and good solubility in THF.

In order to further understand the impacts of reduced crystallinity of BM-ClEH on aggregation strength, UV-vis absorption measurements were conducted. It is found that the pronounced *O-1* peak in B3T-P[53] (a reported SMD similar to BM-HEH with *n*-butyl on Rh) absorption becomes weak in the absorption of BM-HEH as indicated by the broader half-width of the *O-1* peak (Fig. 1c), which illustrates that alkyl chain change of substituent on Rh group from *n*-butyl to 2-ethylhexyl will significantly reduce aggregation strength of SMD. Based on this observation, we are more concerned about how much aggregation property of BM-Cl can be preserved in BM-ClEH. So, temperature-variable UV-vis absorption spectra were measured (Supplementary Fig. 4). It can be observed that BM-Cl and BM-ClEH exhibit nearly identical absorption spectra in CF and THF solution at different temperature with same intensities of *O-0* and *O-1* peaks (Fig. 1d and Supplementary Fig. 4a), which indicates that BM-ClEH can perfectly inherit the strong aggregation property of BM-Cl, attributing to strong intramolecular Cl-S interaction.

Further insights into the effects of different solvents on pre-aggregation state of involved donor and acceptor materials were studied. From CF to THF, the *O-0* peak of BO-4Cl undergoes a significant blue shift of ~ 22 nm (Fig. 1e), suggesting that pre-aggregation state of BO-4Cl will become weaker in THF solution. Moreover, the intensity of *O-0* peak of BO-4Cl in CF decreases very slightly with gradual increase of temperature (Supplementary Fig. 4d). However, when coming into THF, BO-4Cl undergoes a sharp drop of *O-0* peak with increasing temperature (Supplementary Fig. 4h). We speculated that the reason why pre-aggregation of BO-4Cl is greatly disrupted in THF atmosphere is because the bulky THF

molecule can penetrate into molecular packing of BO-4Cl under high temperature-induced thermal motion and thus destroy molecular aggregation, making BO-4Cl molecules become more freedom and disorder. Such situation also takes place in THF solution of weakly aggregated BM-HEH, but its degree is not as severe as that of BO-4Cl (Supplementary Fig. 4e, i), which is very likely caused by a relatively stronger intermolecular interaction between BM-HEH than BO-4Cl. It is worth mentioning that the fine structures of absorption spectra of BO-4Cl film spin-coated from THF is same to that spin-coated from CF (Fig. 1e), indicating molecular packing of BO-4Cl can be well self-healed after THF molecule are removed. However, BM-HEH cannot do like this so that BM-HEH film formed from THF shows a 10 nm bule-shift compared to the one formed from CF (Fig. 1c). Similarly, the *O-0* peak of BM-ClEH is also destroyed in THF but the strong *O-1* peak can remain unchanged even under high temperature (Fig. 1d and Supplementary Fig. 4g), and corresponding film coated from THF even slightly red-shifts relative to the one coated from CF. The above results inform us that strong aggregation property of BM-ClEH is able to well maintain pre-aggregation state in THF, which is conducive to obtain a good film-forming morphology.

## Molecular rigidity

To gain deeper understanding into mechanisms underlying the enhancement of aggregation property by intramolecular Cl-S interaction, temperature-variable NMR measurements of three SMDs were performed by measuring every 4 °C within range from 20 °C to 56 °C. [1]H NMR spectra under different temperature and chemical shift changes of different H atoms varying along with temperature changes are shown in Fig. 2, Supplementary Figs. 5, 6 and Supplementary Tables 1–3. Figure 2a displays the locations of different H atoms on BM-Cl, BM-ClEH and BM-HEH named from Ha to Hh. It is acknowledged that the vibration and rotation of C-H bond will be strengthened to achieve a new molecular equilibrium conformation when temperature of surrounding environment increases, which will result in a slight change in chemical environment of H atom. However, the movement of C-H bond will be inhibited when molecular structure becomes more rigid, so the overall rigidity of the molecule can be judged by observing

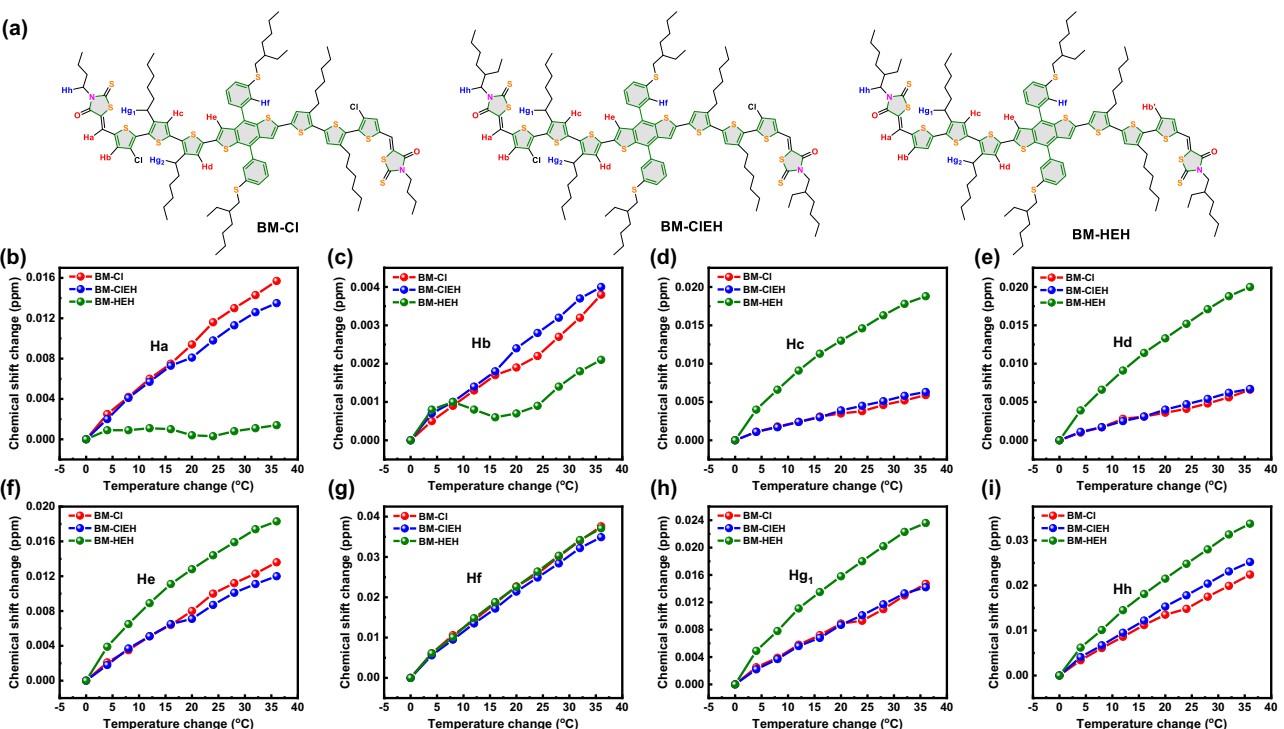

**Fig. 2 | Temperature-variable NMR. a** Locations of different H atoms on BM-Cl, BM-ClEH and BM-HEH. Chemical shift changes varying along with temperature changes for (**b**) Ha; (**c**) Hb; (**d**) Hc; (**e**) Hd; (**f**) He; (**g**) Hf; (**h**) $Hg_1$ and (**i**) Hh.

chemical shift change rate of H atom with temperature. It is not difficult to find that chemical shift change rate of H atom varying along with temperature on alkyl chains are significantly faster than these on molecular backbone (Supplementary Figs. 6a–c), especially Hf on 2D-phenyl side chain (Fig. 2g), which is consistent with our previous cognition that flexible alkyl chains will reduce molecular rigidity. However, introducing intramolecular Cl-S non-covalent interaction not only greatly reduce chemical shift change rate *vs* temperature of Hc, Hd and He on SMD's skeleton (Fig. 2d–f) but also is able to suppress such a rate of H atoms ($Hg_1$, $Hg_2$ and Hh) on methylene linked to thiophene and Rh groups (Fig. 2h–i and Supplementary Fig. 6d), demonstrating a significant increase in molecular rigidity of BM-Cl and BM-ClEH relative to BM-HEH. It is worth noting that Ha and Hb atoms on BM-HEH seems to be steadier than those on BM-Cl and BM-ClEH (Fig. 2a, b), which is most likely due that electron-deficient Cl atom reduces the electron-rich ability of S atom of 3-chlorothiophene and thereby weakening S-S non-covalent interaction between thiophene and Rh groups. This phenomenon also further indicates the great effect of non-covalent interactions on the improvement of molecular rigidity. Overall, Cl-S non-covalent interaction within BM-Cl and BM-ClEH enhances molecular rigidity as a result to obtain strong aggregation property.

**Films morphology, mobility and overall interactions**
Changes in crystallinity and pre-aggregation state of involved materials and differences of processing solvents in vapor pressure will have impacts on thermodynamic and kinetic factors of transition from solution to solid film, thus resulting in different morphology of films. Therefore, pure films of three SMDs and acceptor BO-4Cl coated from CF and/or THF solvents (i.e., BM-Cl (CF), BM-ClEH (CF), BM-ClEH (THF), BM-HEH (THF), BO-4Cl (CF) and BO-4Cl (THF)) were characterized by employing grazing incidence wide-angle X-ray diffraction (GIWAXS) (Supplementary Fig. 7 and Fig. 3a–b) and space charge limited current (SCLC) techniques (Supplementary Fig. 8 and Fig. 3c, d) to observe molecular orientation and packing as well as charge

transport abilities. Data related to *d*-spacings and coherence lengths (CLs) were extracted from GIWAXS cut-line profiles and listed in Supplementary Table 4. As shown in GIWAXS 2D-patterns that BM-Cl (CF), BM-ClEH (CF)/(THF) and BM-HEH (THF) films all displays well-define "edge-on" orientations with strong (100) lamellar diffraction peak in out-of-plane (OOP) direction (*d*-spacings/CLs are 19.5/99.8, 19.3/67.2, 19.5/79.7 and 18.6/97.7 Å, respectively) and weak (010) π-π stacking diffraction peak in in-plane (IP) direction (*d*-spacings/CLs are 3.66/15.1, 3.87/4.3, 3.85/5.5 and 3.86/7.6 Å, respectively). Alkyl chain extension on Rh group of SMDs does not affect lamellar packing too much but is the culprit for weakening π-π stacking (enlarging *d*-spacing and decreasing CLs), which remarkably reduces the hole mobilities of SMDs (Fig. 3c). Although molecular packing of BM-ClEH (THF) is slightly weaker than that of BM-HEH (THF), its hole transport ability is significantly stronger than that of BM-HEH (THF), which is probably attributed to Cl-S non-covalent interaction to improve molecular rigidity and planarity. For BO-4Cl, it adopts a "face-on" orientation, but its π-π stacking of film spin-coated from THF is slightly weaker than the one from CF and thus a drop in electron mobility. We found that for molecules (BM-HEH and BO-4Cl) that the pre-aggregated sate (weak aggregation) is easily destroyed in THF solution, their *d*-spacings and CLs of lamellar packing in films will be obviously improved compared to molecules (BM-Cl and BM-ClEH) that the pre-aggregated state (strong aggregation) are not easily destroyed in THF solution. This phenomenon well explains what we said before that THF can penetrate between molecules because they can induce an aggregation process of molecules when THF is volatilized, just like the role of volatile additives[21,54,55].

In order to illustrate overall interaction strength of different films, we measured the thermal transition by taking advantage of red-shifts of UV-vis absorption spectrum that occurs after thermally annealing as-cast films (Supplementary Fig. 9). Generally, if molecule itself interacts stronger, its thermal-induced motion will be slower, which can be quantified by deviation metric ($DM_T$) that is the sum of squared deviation in the absorbance between as-cast and annealed

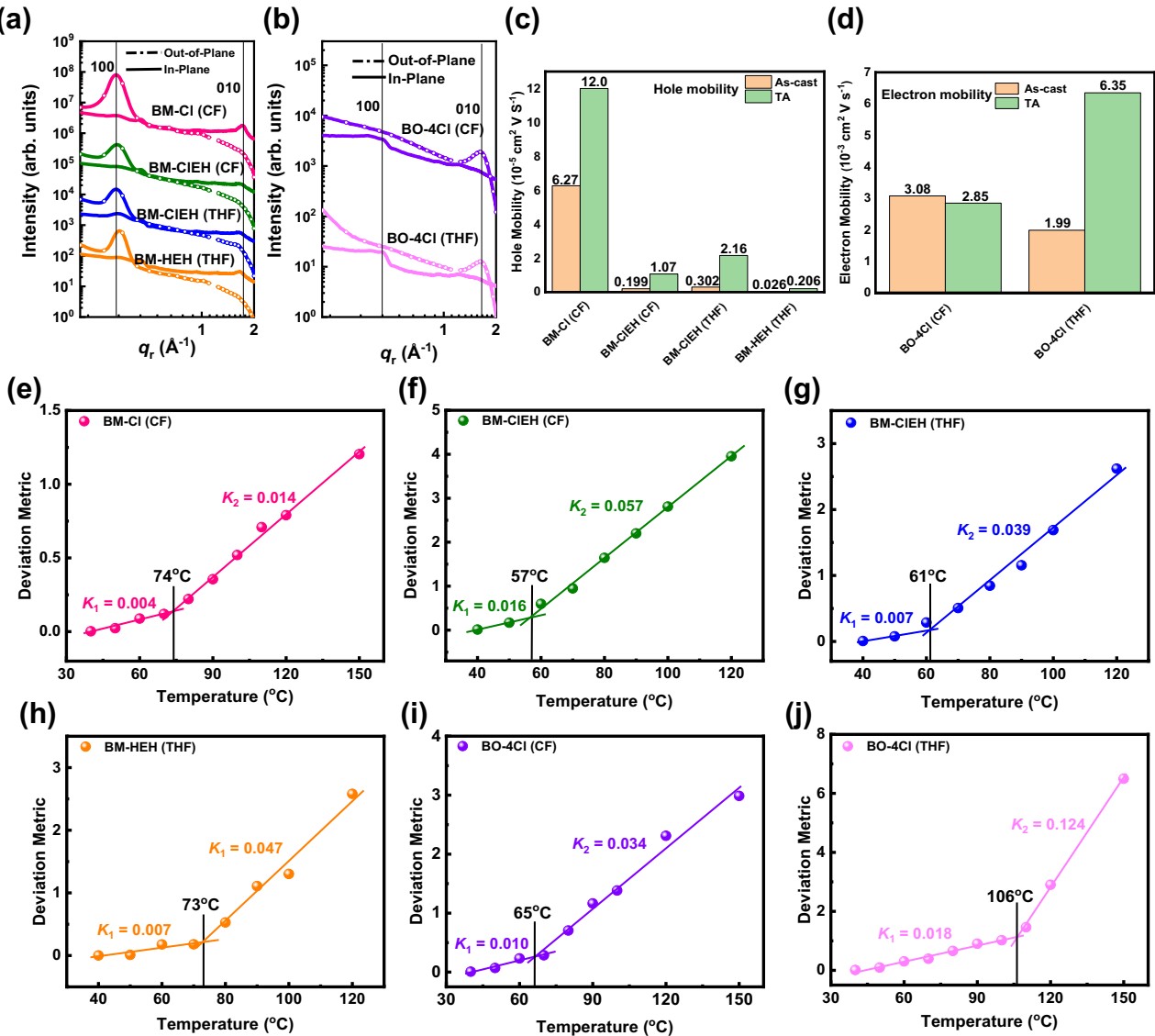

**Fig. 3 | π-π stacking, mobility and overall interactions for neat films. a** GIWAXS cut-line profiles for BM-Cl (CF), BM-ClEH (CF), BM-ClEH (THF) and BM-HEH (THF) films. **b** GIWAXS cut-line profiles for BO-4Cl (CF) and BO-4Cl (THF) films. **c** Hole mobilities of BM-Cl (CF), BM-ClEH (CF), BM-ClEH (THF) and BM-HEH (THF) films. **d** Electron mobilities of BO-4Cl (CF) and BO-4Cl (THF) films. Deviation metric of redshifts measured from temperature-variable UV-vis absorption spectra relative to temperature for (**e**) BM-Cl (CF); (**f**) BM-ClEH (CF); (**g**) BM-ClEH (THF); (**h**) BM-HEH (THF), (**i**) BO-4Cl (CF) and (**j**) BO-4Cl (THF) films.

films (Fig. 3e–j)[56]:

$$DM_T = \sum_{\lambda_{\min}}^{\lambda_{\max}} [A_{RT}(\lambda) - A_T(\lambda)]^2 \qquad (1)$$

where $\lambda_{\min}$ and $\lambda_{\max}$ are the lower and upper bounds of optical sweep, $A_{RT}(\lambda)$ and $A_T(\lambda)$ are normalized absorption intensities of as-cast and annealed films, respectively. In general, when change rate of $DM_T$ value with temperature becomes faster, it indicates that red-shift of absorption spectra under annealing is larger, corresponding to thermal motion of molecules is more obvious, that is weaker interaction, and vice versa. As shown in Fig. 3e, the $DM_T$ value of BM-Cl (CF) film is smallest accompanied by slowest change rate with temperature no matter before ($K_1$) or after ($K_2$) transition temperature ($T_g = 74\,^\circ\text{C}$), suggesting overall intermolecular interaction between BM-Cl molecules within film is the strongest. The $DM_T$ value of BM-ClEH (CF) film is almost three times that of BM-Cl (CF) with $T_g$ droping

to $57\,^\circ\text{C}$ and $K_1$, $K_2$ greatly increased (Fig. 3f). Importantly, when changing CF with THF as processing solvent, the interaction between BM-ClEH molecules will be strengthened as $T_g$ raises and $DM_T$ value, $K_1$ and $K_2$ decrease (Fig. 3g). BM-HEH (THF) and BO-4Cl (THF) films show high $T_g$, most likely due to increased lamellar packing as suggested by GIWAXS measurement. Moreover, the stronger molecular rigidity causes BM-ClEH (THF) to exhibit slower $K_2$ after $T_g$ than that of BM-HEH (THF). It is worth noting that BO-4Cl (THF) film undergoes largest thermal motion with $MD_T$ value of up to 7 and $K_2$ of 0.124 (Fig. 3j), which is conducive for BO-4Cl (THF) film to obtain better and more ordered arrangement after TA, thus twice times higher electron mobility is achieved after annealing (Fig. 3d). These results tell us that effect of intermolecular non-covalent Cl-S interactions together with THF processing can enhance the overall strength of interaction between material molecules.

When coming into blend films, molecular aggregation behaviors and thermal motion will become more complex. The absorption spectra of blend films under different TA temperature and solvent

vapor annealing (SVA) time were measured (Supplementary Fig. 10). It can be clearly observed that the large addition of BO-4Cl will partially destroy aggregation of SMDs. However, as TA temperature increases, the *O-O* peak gradually appears and intensifies in BM-Cl:BO-4Cl and BM-ClEH:BO-4Cl films, which can be probably attributed to amorphous components of SMD in blend film are inability to fully revert to its originally strong aggregates but leads to formation of some weak aggregates under TA action. It should be noted that the stronger *O-O* and *O-1* peaks of BM-Cl:BO-4Cl are derived from a stronger crystallinity of BM-Cl than BM-ClEH though a faster rate of thermal motion of BM-ClEH in CF-casted film. BM-ClEH:BO-4Cl (THF) seems to be more sensitive to TA treatment and achieved stronger *O-O* and *O-1* peaks than that of BM-ClEH:BO-4Cl (CF) due to a more significant molecule motion by thermal of BO-4Cl in THF coated film than in CF coated film, which determines a larger $DM_T$ values of BM-ClEH:BO-4Cl (THF) than BM-ClEH:BO-4Cl (CF) (Supplementary Fig. 11). Likewise, the smaller $DM_T$ value of BM-Cl:BO-4Cl (CF) than that of BM-ClEH:BO-4Cl (CF) can be due to weaker thermal motion of BM-Cl. It seems that the rapid thermal motion of BO-4Cl in turn accelerates the motion of BM-ClEH to facilitate better aggregates in THF coated blend films. SVA is able to give molecules more free movement space in blend films and quickly make donor and acceptor reach a good aggregation state. Such large freedom of molecule motion is beneficial for BM-Cl or BM-ClEH to form strong aggregates compared to TA-treated blend films. However, for BM-HEH, neither TA nor SVA can recover its aggregation state as same as in pure film, indicating that it is completely an amorphous state in the blend films.

Photoluminescence spectra (PL) of BM-Cl (CF), BM-ClEH (CF), BM-ClEH (THF), BM-HEH (THF), BO-4Cl (CF) and BO-4Cl (THF) neat films without/with TA or SVA were measured. As shown in Supplementary Fig. 12, TA and SVA treatments are able to improve molecular aggregation and/or ordering to induce PL emission enhancements (for SMDs) or red-shifts (for BO-4Cl). On the whole, TA effect is more obvious than SVA which is beneficial to achieve better film morphology. Band gaps ($E_g$s) of investigated materials are determined by calculating the intersection of absorption and PL spectra (Supplementary Fig. 13), which are 1.831, 1.866, 1.857, 1.851, 1.439, and 1.431 eV, respectively. In addition, ultraviolet photoelectron spectroscopy (UPS) measurements were performed in order to accurately probe energy levels of SMDs and SMA (Supplementary Fig. 14). The Fermi levels ($E_F$) and valence bands (VB) of above films were fitted to be −4.34/0.55, −4.56/0.49, −4.50/0.56, −4.52/0.50, −4.62/0.98 and −4.45/1.12 eV, respectively, corresponding to highest occupied molecular orbital energy levels ($E_{HOMO}$) of −4.98, −5.05, −5.06, −5.02, −5.60, −5.57 eV, respectively. The lowest unoccupied molecular orbital energy levels ($E_{LUMO}$) were also obtained as −3.06, −3.18, −3.20, −3.17, −4.16, and −4.14 eV, respectively, according $E_{LUMO} = E_{HOMO} + E_g$. Sufficient energy levels offsets between $E_{HOMO}$s or $E_{LUMO}$s of SMDs and SMA can effectively ensure exciton transfer. Interestingly, the Fermi level of THF-processed films elevates relative to CF-processed one and VB widens, especially for BO-4Cl, which is likely to facilitate hole transfer and transport. PL spectra of BM-Cl:BO-4Cl (CF), BM-ClEH:BO-4Cl (CF), BM-ClEH:BO-4Cl (THF) and BM-HEH:BO-4Cl (THF) blend films exhibited significant quenching with efficiencies reach more than 90% after TA and SVA, indicating good charge transfer from SMA to SMDs (Supplementary Fig. 15).

## Photovoltaic performance of THF-processed ASM-OSCs
ASM-OSCs based on a device structure of indium tin oxide (ITO)/ poly(3,4-ethylenedioxythiophene):poly(styrenesulfonate)(PEDOT:PSS)/SMD:BO-4Cl/ poly(9,9-bis(3'-(N,N-dimethyl)-N-ethylammoinium-propyl−2,7-fluorene)-alt-2,7-(9,9-dioctylfluorene))dibromide:Melamine (PFN-Br:MA)[13,17]/Ag were fabricated with details available in Supplementary Information. We tried TA and SVA treatments to optimize device performance. Corresponding photovoltaic

parameters were summarized in Supplementary Tables 6−9 with best-performing current density-voltage (*J-V*) curves shown in Fig. 4 and Supplementary Fig. 16. BM-Cl:BO-4Cl active layer is not suitable for THF-processing due to poor solubility of BM-Cl in THF. Its CF-processed devices achieved 14.3% and 15.4% PCE under conditions of TA (150 °C) and SVA (chlorobenzene, 80 s), respectively, higher than those devices based on BM-ClEH:BO-4Cl under similar conditions (13.6% for TA and 13.7% for SVA) (Table 1). The $J_{SC}$ reduction of devices based on BM-ClEH:BO-4Cl (CF) can be attributed to decline of crystallization performance of BM-ClEH. But this situation has been greatly improved when THF is used to spin-coat BM-ClEH:BO-4Cl active layer and then TA post-treatment (100 °C). As a result, a PCE of 15.0% is achieved for non-halogen solvent processed ASM-OSCs (Fig. 4d). It is worth mentioning that TA is more applicable than SVA for large-scale production because TA is able to overcome poor repeatability of SVA in device preparation. We counted photovoltaic parameters of 20 cells in one batch for each device based on BM-Cl:BO-4Cl (CF), BM-ClEH:BO-4Cl (CF) and BM-ClEH:BO-4Cl (THF) under as-cast, TA and SVA conditions, which are shown in Fig. 4e and Supplementary Fig. 17. TA-treated devices do show better device reproducibility than SVA-treated ones, as shown in the better repeatability of three device parameters. Therefore, BM-ClEH:BO-4Cl (THF) device is of great significance for commercial application of OSCs in the near future. However, OSCs devices based on BM-HEH:BO-4Cl processed with THF have almost no PCE (less than 1%) regardless of TA or SVA, which strongly illustrates the importance of strong aggregation property in achieving highly-efficient non-halogen solvent-processed ASM-OSCs.

To further demonstrate the potential of BM-ClEH in different non-halogen solvents and compatibility with other acceptor materials, ASM-OSCs with carbon disulfide (CS₂) as processing solvent or BTP-eC9 and L8-BO as electron acceptors were fabricated. Optimized *J-V* curves of corresponding devices were displayed in Supplementary Fig. 18. with photovoltaic parameters summarized in Supplementary Table 11. As a result, ASM-OSCs based on BM-ClEH:BO-4Cl processed with CS₂ can achieve close to 11% PCE along with a 71.2% FF under TA post-treatment at 120 °C, which is a relatively high efficiency for such low boiling point (46.2 °C) solvent processing and a rare case for OSCs processed with CS₂. When altering acceptor materials, ASM-OSCs based on BM-ClEH:BTP-eC9 and BM-ClEH:L8-BO fabricated with THF gained 12.6% and 13.1% PCE, respectively, indicating a good morphology compatibility of BM-ClEH with different acceptors in non-halogen solvent processing.

A series of device characterizations are carried out. BM-Cl:BO-4Cl (CF) devices achieved highest external quantum efficiency (EQE) response (Fig. 4c) under both TA and SVA conditions compared to other devices due to excellent exciton dissociation efficiency (~98%) and high electron/hole mobility ($\mu_e/\mu_h$) (Supplementary Figs. 19, 20 and Supplementary Tables 12, 13). While EQE intensities of devices based on BM-ClEH:BO-4Cl processed with CF or THF performed differently (50−80%) for TA and SVA treatments. The combination of THF-processing and TA treatment enables its devices to achieve very balanced carrier mobility ($\mu_e/\mu_h = 1.26$) because of significantly improved hole transport ability of active layer. Other combinations of conditions lead to lower EQEs for BM-ClEH:BO-4Cl-based devices either due to poor exciton dissociation or imbalanced carrier transport. Moreover, devices based on BM-HEH:BO-4Cl (THF) show no more than 10% EQE intensity under different conditions, the extremely poor exciton dissociation/collection efficiencies and low carrier mobility are the main reasons for inefficient photoelectric conversion process. With crystallization property of SMDs decreasing and processing solvent changing, electron mobilities of blend films will be greatly affected (Fig. 4f) due to reduced lamellar packing and π-π stacking (Supplementary Table 14). The integrated $J_{SC}$s of four optimal devices were calculated for each condition and listed in Table 1, which is in good agreement with the tested results within 3% error.

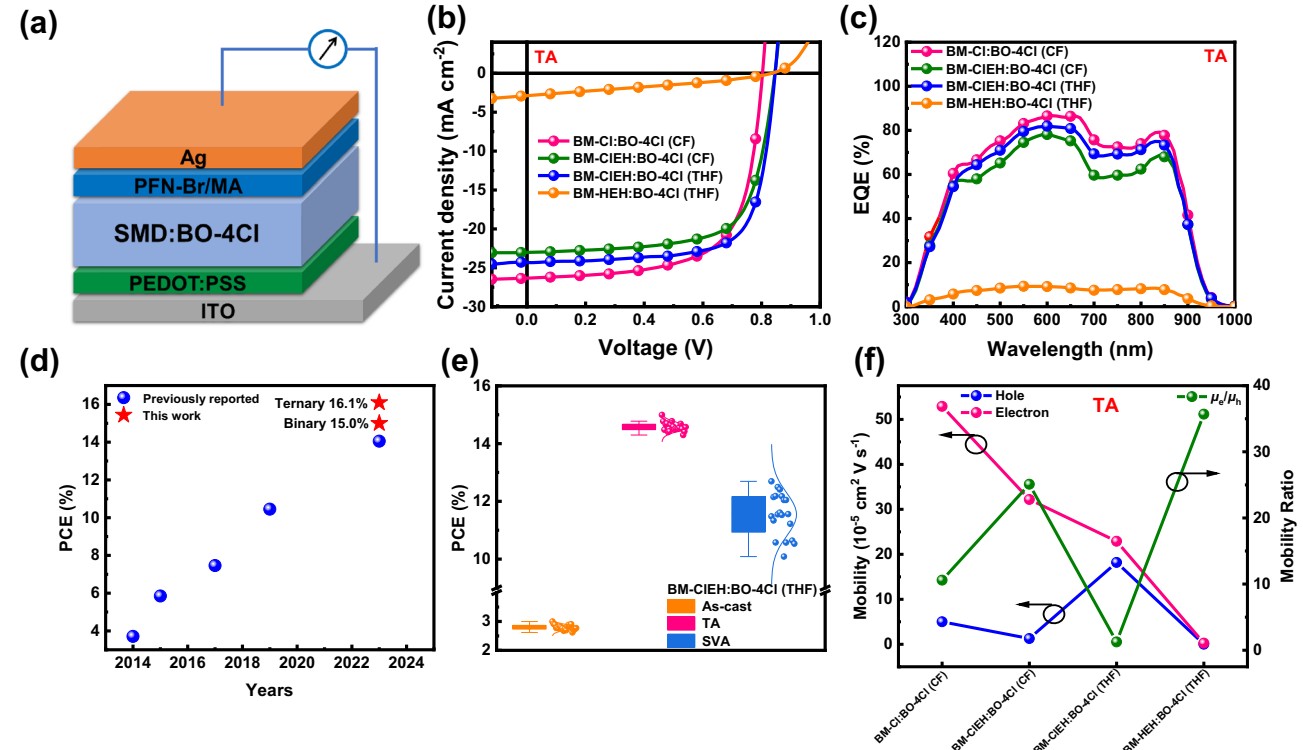

**Fig. 4 | Device structure and performance of ASM-OSCs. a** Device structure. **b** *J-V* curves for best-performing non-halogen solvent processed devices under TA. **c** Corresponding EQE spectra. **d** Comparison of ASM-OSCs processed by non-halogen solvents. **e** Device reproducibility of BM-ClEH:BO-4Cl (THF) devices under as-cast, TA and SVA conditions. **f** Hole/electron mobilities for BM-Cl:BO-4Cl (CF), BM-ClEH:BO-4Cl (CF), BM-ClEH:BO-4Cl (THF) and BM-HEH:BO-4Cl (THF) blend films under TA.

**Table 1 | Photovoltaic performance of non-halogen solvent processed ASM-OSCs based on BM-Cl:BO-4Cl (CF), BM-ClEH:BO-4Cl (CF), BM-ClEH:BO-4Cl (THF) and BM-HEH:BO-4Cl (THF) under different conditions**

| Active layer | Treatment | $V_{oc}$ (V) | $J_{sc}$ (mA cm⁻²) | $J_{sc}$[a] (mA cm⁻²) | FF (%) | PCE[b] (%) |
|---|---|---|---|---|---|---|
| BM-Cl:BO-4Cl (CF) | As-cast | 0.862 | 14.84 | 14.40 | 39.8 | 5.1 (4.8 ± 0.1) |
| | TA (150 °C) | 0.803 | 26.34 | 25.72 | 67.4 | 14.3 (13.8 ± 0.2) |
| | SVA (80 s)[c] | 0.806 | 26.23 | 25.62 | 72.9 | 15.4 (14.3 ± 0.5) |
| BM-ClEH:BO-4Cl (CF) | As-cast | 0.909 | 7.47 | 7.93 | 34.8 | 2.4 (2.1 ± 0.1) |
| | TA (120 °C) | 0.845 | 23.03 | 22.62 | 69.8 | 13.6 (13.4 ± 0.1) |
| | SVA (80 s)[c] | 0.835 | 22.26 | 21.95 | 73.6 | 13.7 (12.8 ± 0.6) |
| BM-ClEH:BO-4Cl (THF) | As-cast | 0.910 | 8.32 | 8.66 | 39.6 | 3.0 (2.8 ± 0.1) |
| | TA (100 °C) | 0.846 | 24.38 | 24.22 | 72.7 | 15.0 (14.6 ± 0.1) |
| | SVA (80 s)[c] | 0.838 | 20.36 | 19.53 | 74.4 | 12.7 (11.5 ± 0.7) |
| BM-HEH:BO-4Cl (THF) | As-cast | 0.795 | 0.47 | 0.49 | 24.9 | 0.1 |
| | TA (100 °C) | 0.834 | 2.87 | 2.68 | 30.5 | 0.7 |
| | SVA (80 s)[c] | 0.574 | 0.54 | 0.45 | 26.1 | 0.1 |

[a]Integrated from EQE spectra.
[b]Inside the brackets are the average of 20 devices.
[c]Using chlorobenzene.

## Active layer morphology study

The difference in photovoltaic performance of the above four ASM-OSCs devices is inseparable from nanomorphology of active layers formed under different processing conditions. To figure out this, systematic morphology measurements including transmission electron microscope (TEM), atomic force microscope (AFM), GIWAXS and grazing incidence small-angle X-ray scattering (GISAXS) were performed with results displayed in Fig. 5, Supplementary Figs. 21–28 and Supplementary Tables 14, 15. It can be observed that CF-spin coated active layers based on BM-Cl and BM-ClEH both exhibit uniform and smooth surfaces with small mean square error of roughness (RMS)

around 1 nm (Fig. 5a–f), which may result from good miscibility of blend solution due to excellent solubility of CF to BM-Cl, BM-ClEH and BO-4Cl. Compared with BM-ClEH:BO-4Cl (CF) film, BM-Cl:BO-4Cl (CF) film appears many porous structures likely due to stronger crystallinity of BM-Cl (Fig. 5a), which leads to a lower hole mobility of BM-Cl:BO-4Cl (CF) as-cast film although a stronger intermolecular lamellar packing and π-π stacking (Supplementary Tables 13, 14, *d*-spacing/CLs in IP and OOP direction are 17.7/16.8 and 3.89/9.7 Å for BM-Cl:BO-4Cl (CF), 18.2/10.8 and 3.95/6.4 Å for BM-ClEH:BO-4Cl (CF), respectively). After TA and SVA treatment, RMS values were decreased and donor/acceptor composition became more uniform for both BM-Cl:BO-4Cl (CF) and

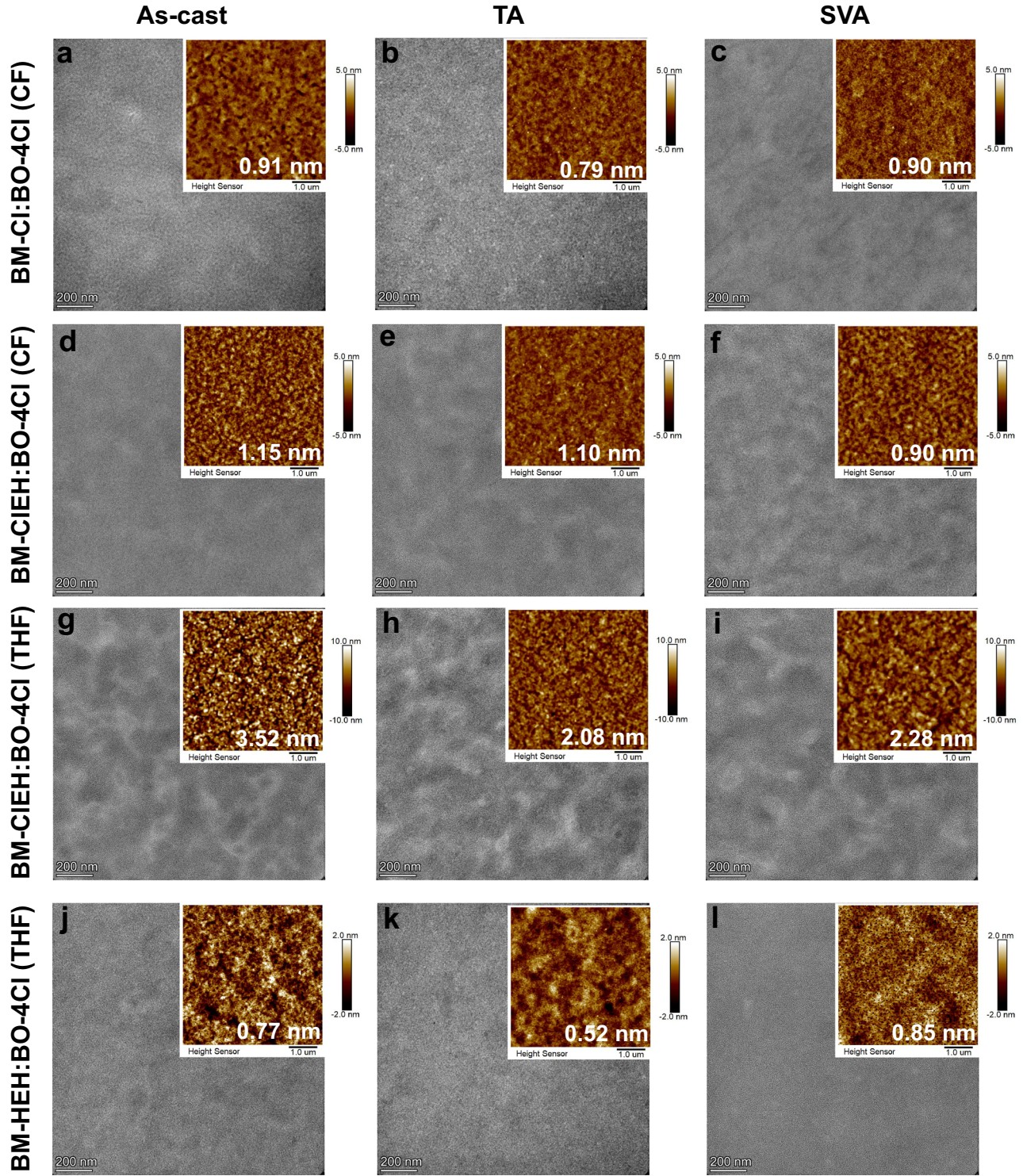

**Fig. 5 | Morphology study of different active layers.** TEM and AFM images for BM-Cl:BO-4Cl (CF) films under (**a**) as-cast, (**b**) TA and (**c**) SVA conditions; BM-ClEH:BO-4Cl (CF) films under (**d**) as-cast, (**e**) TA and (**f**) SVA conditions; BM-ClEH:BO-4Cl (THF) films under (**g**) as-cast, (**h**) TA and (**i**) SVA conditions; BM-HEH:BO-4Cl (THF) films under (**j**) as-cast, (**k**) TA and (**l**) SVA conditions.

BM-ClEH:BO-4Cl (CF) layers. Although TA and SVA show little effects on the domain sizes of donor and acceptor phases (Supplementary Table 15), but can significantly enhance intensity and ordering of π-π stacking, which improves carrier mobility and balanced transport, thus in turn promoting FF and $J_{SC}$. Slightly lower PCE of optimal AMS-OSCs based on BM-ClEH:BO-4Cl (CF) than that based on BM-Cl:BO-4Cl (CF) can be attributed to a weaker π-π stacking of BM-ClEH:BO-4Cl (CF)

layer (*d*-spacing/CLs of BM-Cl:BO-4Cl (CF) and BM-ClEH:BO-4Cl (CF) are 3.73/16.9, 3.82/13.1 Å after TA and 3.67/13.7, 3.85/10.1 Å after SVA, respectively) that leads to lower hole/electron mobilities and thus lower $J_{SC}$.

BM-ClEH:BO-4Cl film coated from THF exhibits a three times RMS value than that film coated from CF (3.52 vs 1.15 nm) with a striking difference in TEM images that comes from uneven electron scatter of

different areas in observation range (Fig. 5d–i). We speculated that this large morphology difference is probably caused by the slower volatility of THF that promoting better crystallization of BM-ClEH as we demonstrated above that the interaction between BM-ClEH in film coated from THF is stronger than that coated from CF. Moreover, enhanced CLs for both lamellar and π-π packing in IP and OOP directions and larger phase scale of BM-ClEH in BM-ClEH:BO-4Cl (THF) film can also illustrate this phenomenon. TA treatment not only significantly reduces the surface roughness (from 3.52 to 2.08 nm) of BM-ClEH:BO-4Cl (THF) film but also makes the distribution of donor/acceptor components more uniform with larger and more ordered donor/acceptor crystals. As a result, the hole and electron mobilities of BM-ClEH:BO-4Cl (THF) film are simultaneously enhanced by a similar magnitude, achieving a more balanced carrier transport performance than that of BM-ClEH:BO-4Cl (CF) film. In contrast, interactions between BO-4Cl in film casted from THF is weaker than that in film casted from CF. As a result, SVA treatment will allow an easier movement of BO-4Cl molecule in BM-ClEH:BO-4Cl (THF) film than in BM-ClEH:BO-4Cl (CF) film, which causes BO-4Cl to form larger domain (sizes of BO-4Cl phase is 34.4 nm for BM-ClEH:BO-4Cl (THF) and 26.9 nm for BM-ClEH:BO-4Cl (CF)) within a short period of time. It can be clearly seen from the AFM and TEM images that SVA causes a larger phase separation of BM-ClEH:BO-4Cl (THF) film, which reduces the exciton dissociation efficiency. In addition, excessively large acceptor phase is not conducive to the arrival of excitons to the D/A interface. These factors lead to a lower $J_{SC}$ in BM-ClEH:BO-4Cl (THF)-based devices than in BM-ClEH:BO-4Cl (CF)-based devices under SVA condition. However, the morphology of BM-HEH:BO-4Cl (THF) film is greatly different from that of BM-ClEH:BO-4Cl (TFH), showing very smooth surface but no obvious phase separation and nano-interpenetrating network structures as well as many pinhole defects even though after TA and SVA treatment (Fig. 5j–l). Moreover, although the domain sizes of donor and acceptor phases of BM-HEH:BO-4Cl (THF) are close to ideal, the ordering of its π-π stacking is severely inhibited due to amorphous crystalline nature of BM-HEH. These unfavorable morphological factors directly cause BM-HEH:BO-4Cl (THF) active layer to almost lose the capabilities of exciton dissociation and carrier transport. The contrast in morphology of BM-ClEH:BO-4Cl (THF) and BM-HEH:BO-4Cl (THF) films strongly illustrates the impacts of intramolecular Cl-S non-covalent interaction on favorable morphology formation of active layer based on non-halogen solvent processing.

Combining all of the above characterizations and photovoltaic performance of four ASM-OSCs, a further understanding on film morphology evolution driven by TA or SVA can be portrayed as Fig. 6. Chemical structures of material molecule itself including alkyl chain and end group and processing solvent co-determine the molecular pre-aggregation, which directly corresponds to initial morphology of as-cast film and developing direction of non-equilibrium state after different annealing method. BM-Cl molecules can aggregate/self-assemble strongly in solution state (CF solvent) to form a partially separated donor-acceptor phase (Fig. 6a), which is responsible for the best as-cast PCE of ASM-OSCs based on BM-Cl:BO-4Cl (CF). TA post-treatment enables BM-Cl:BO-4Cl (CF) active layer to achieve a more bridged donor phase due to reactivated material de-mixing and molecule diffusion (Fig. 6a). In contrast, SVA brings a more complicated process of phase reformation including crystal dissolution, recrystallization, phase separation and purification[57], and finally realizes an even better connected donor networks with protected D/A interface area, which is easily obtained by PDs due to their strong pre-aggregation property. Parallelly, BM-ClEH with prolonged alkyl chain on Rh group exhibits a weaker crystallinity, which determines crystallization rate and process of BM-ClEH:BO-4Cl (CF) to make as-cast film has less preformed donor-rich phase but more amorphous D/A intermixing clusters although strong pre-aggregation strength of BM-

ClEH is not much different from that of BM-Cl (Fig. 6b). As a result, both TA and SVA can only leads to limitedly connected donor phase, so there exist no clear PCE difference between two post-treatments. As for THF as-casted BM-ClEH:BO-4Cl film, significant pre-aggregation of BM-ClEH can effectively prevent THF molecules from destroying donor clusters that has unique property of pre-connection, which provides a most potential morphology beginning point of achieving interconnecting fibrillar network, yet temporarily effective D/A interface for exciton splitting is not sufficient (Fig. 6c). As a result, SVA process leads to slightly over-aggregated behavior for final morphology due to relatively loose molecular packing of BM-ClEH:BO-4Cl (THF) compared to BM-Cl:BO-4Cl (CF) system, while TA presents a film with properly deposited interpenetrating network. At last, we assign there is no significant morphology change for BM-HEH:BO-4Cl (THF) film before and after post treatments for the lack of strong molecular aggregation ability (Fig. 6d).

## Exciton dissociation and charge recombination

Furthermore, femtosecond-resolved transient absorption spectroscopy (fs-TAS) of neat and blend films were measured (Supplementary Figs. 29–31) to study the dynamics of photo-induced charge carriers by using a suitable excitation fluence (3 μJ cm$^{-2}$ for neat films and 5 μJ cm$^{-2}$ for blend films). The excitation wavelengths selective for SMDs (BM-Cl, BM-ClEH and BM-HEH) and acceptor (BO-4Cl) are determined as 400 nm and 800 nm, respectively, according to their absorption spectra. We choose the fs-TAS spectral line-cuts immediate after excitation (i.e., 0.5–1.0 ps) to represent the singlet excitons that is characterized by the ground state bleach (GSB) signal, as shown in Fig. 7a–d. The maximum GSB for SMDs and acceptor are found to be located at 620–680 nm and 650–690 nm, respectively, thereby used as the corresponding probe ranges in the following investigations. It can be observed that the GSB of neat films after TA and SVA will be red-shifted relative to its as-cast film, especially for films under TA treatment, indicating a stronger intermolecular aggregation, which is completely consistent with the results of GIWAXS tests. In addition, we noted that the negative features around 720 nm is deeper upon TA and SVA. This range is the donor electroabsorption (EA) which is due to free charges formation leading to stark effect then EA features. Additionally, the polarons photo-induced absorption upon exciton dissociations are also expected to closely overlap with the EA on the said range. That means TA- and SVA-induced aggregation is able to enhance exciton dissociation of donor itself[58,59], and this trend is more obvious in BM-ClEH while weakest in BM-HEH. This can be attributed to the influence of amorphous crystalline property of BM-HEH due to lack of intermolecular Cl-S non-covalent interaction, which is mutually corroborated with the conclusions obtained from the morphology tests. Figure 7e–h shows the fs-TAS kinetics of singlet excitons of different neat films of SMDs. The singlet exciton lifetimes for as-cast, TA and SVA films were fitted to be 49 ± 2, 25 ± 1 and 30 ± 1 ps for BM-Cl (CF), 97 ± 15, 71 ± 10 and 67 ± 6 ps for BM-ClEH (CF), 69 ± 4, 52 ± 3 and 62 ± 5 ps for BM-ClEH (THF), and 212 ± 30, 147 ± 11 and 198 ± 30 ps for BM-Cl (CF), respectively. BM-Cl with strongest crystallinity exhibits shortest singlet exciton lifetime, but its exciton diffusion length may be long due to high carrier mobility that supported by ordered π-π stacking (Supplementary Tables 13, 14). TA and SVA slightly shorten the singlet lifetime but enhance π-π stacking intensity and ordering of films to significantly increase carrier mobility, thereby higher exciton diffusion length of films can be obtained after TA and SVA. BM-ClEH (THF) film achieved a balance between molecular crystallinity (i.e., higher carrier mobility) and singlet exciton lifetime compared to BM-ClEH (CF) films, which is beneficial to obtain a higher exciton diffusion length, especially for films under TA treatment. BM-HEH shows longest singlet lifetime that is three times of BM-ClEH (THF). However, its carrier mobility is nearly three orders of magnitude lower than that of BM-ClEH and thereby resulting in a very short exciton diffusion length,

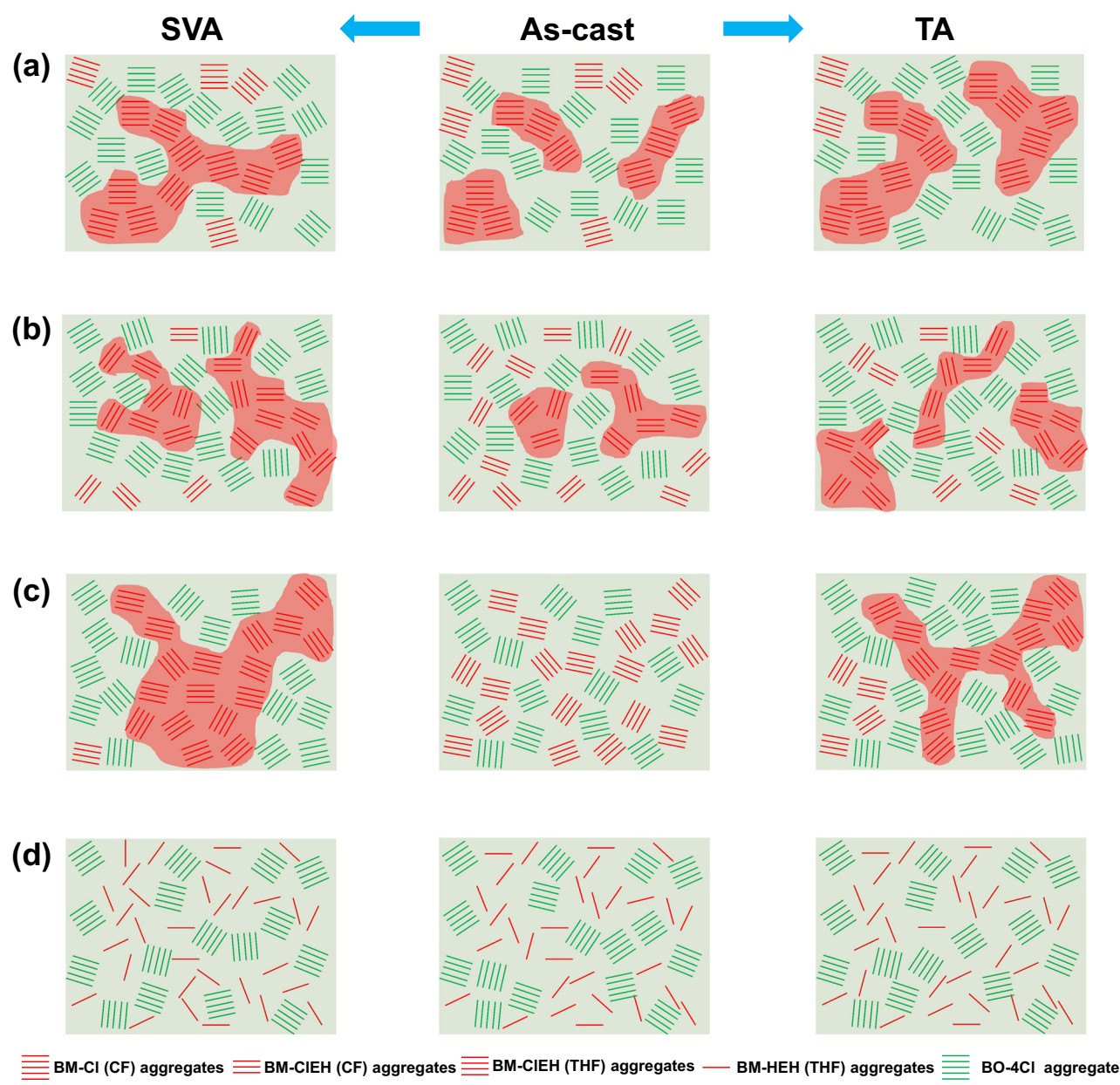

BM-Cl (CF) aggregates   BM-ClEH (CF) aggregates   BM-ClEH (THF) aggregates   BM-HEH (THF) aggregates   BO-4Cl aggregates

**Fig. 6 | Morphology evolution analysis. a** BM-Cl:BO-4Cl (CF). **b** BM-ClEH:BO-4Cl (CF). **c** BM-ClEH:BO-4Cl (THF). **d** BM-HEH:BO-4Cl (THF).

which will lead to photogenerated excitons by donor phase trapped in bulk phase of active layer and unable to reach to D/A interface (the size of donor phase is similar to others suggested by GISAXS). The singlet lifetime of BO-4Cl was also probed (Supplementary Fig. 30), and it was found to be very similar around 25 ps for CF- and THF as-cast films, TA and SVA films, which means that donors' singlet lifetime plays a dominant role on photovoltaic performance of ASM-OSCs in terms of dynamics of photo-induced charge carriers.

As depicted in Supplementary Fig. 31, the fs-TAS of four as-cast, TA and SVA active layers were also measured. Here, the wavelength range of 630–680 nm representing the dynamics of free charges on the basis of hole polarons was selected. As shown in Fig. 7i–l, the rise in kinetics of all active layers related to free charge generation is due to the increasing photobleach from polarons generated upon singlet exciton dissociation. Overall, there are only marginal differences between the rise kinetics for all blends. The higher attainable maximum $\Delta T/T$ signal intensity corresponding to hole polarons GSB range generally indicates a higher free charge population formed. Upon

considering the range (550–600 nm) where the hole polarons photobleach contribution is expected to be more dominant than the singlet GSB (as indicated by the donor and acceptor absorption spectra), the TA and SVA films display a more evident rise of $\Delta T/T$ (5–10 ps) which implies the more efficient exciton dissociation and consistently supporting the higher $J_{SC}$/EQEs of corresponding devices. In terms of hole polarons photobleach decay in the sub-ns range thereby mainly representing the bimolecular losses, it is very evident that TA and SVA were able to prolong the recombination time (or slow free charge recombination). Consequently, much higher FFs are obtained from TA and SVA devices and the minor differences in TA and SVA devices can be justified from the free charge recombination beyond the sub-ns range which is expected to mostly trap-assisted recombination thereby highly domain morphology/mobilities related. This is except BM-HEH:BO-4Cl (THF) film because it technically does not generate significant free charges. So, the kinetics observed from BM-HEH:BO-4Cl (THF) film mainly corresponds to singlet exciton lifetimes which appears unaffected by TA or SVA.

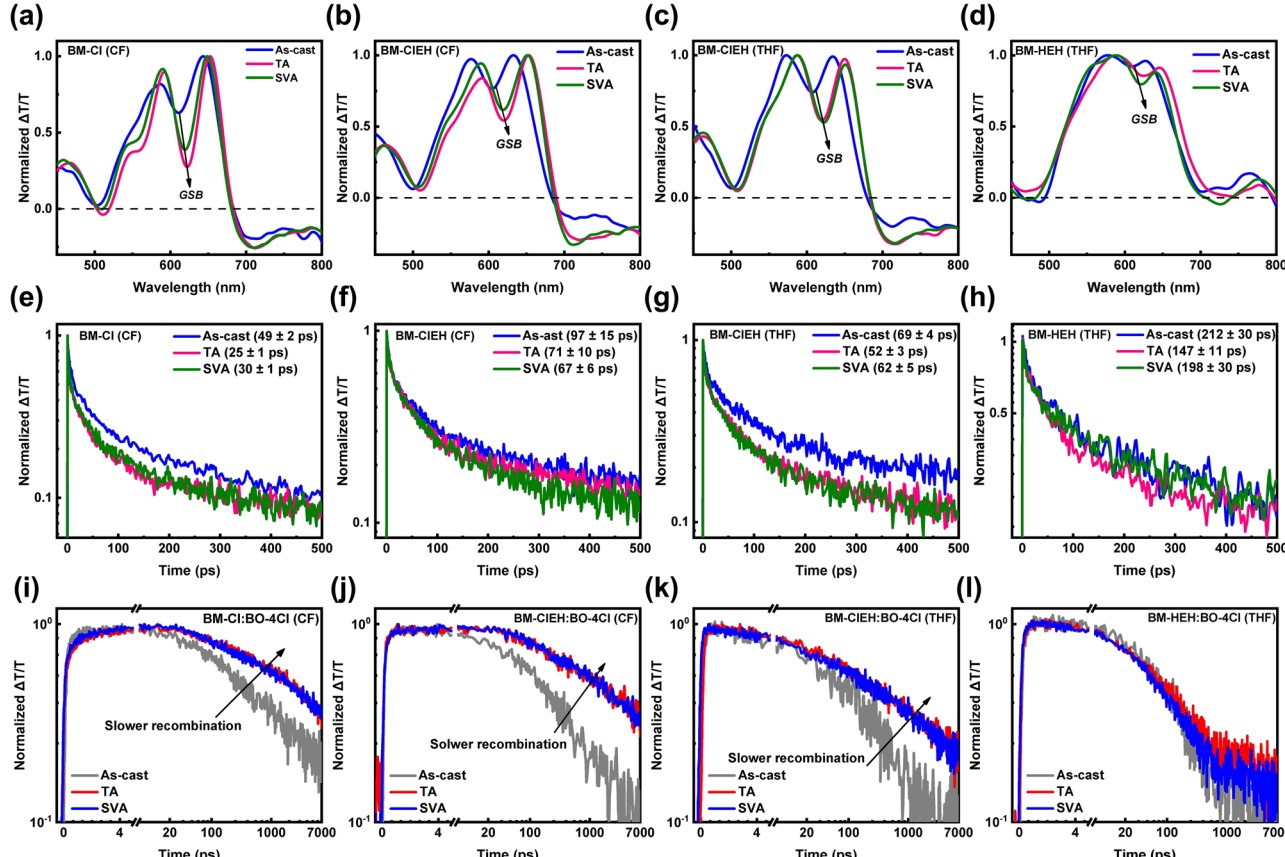

**Fig. 7 | Femtosecond-resolved transient absorption spectroscopy.** fs-TAS spectra presented in terms of ΔT/T. Spectral cuts immediately after excitation (0.5–1 ps) for (**a**) BM-Cl (CF), (**b**) BM-ClEH (CF), (**c**) BM-ClEH (THF) and (**d**) BM-HEH (THF) films under different processing conditions. Corresponding TAS kinetic lifetimes of singlet excitons for (**e**) BM-Cl (CF), (**f**) BM-ClEH (CF), (**g**) BM-ClEH (THF) and (**h**) BM-HEH (THF) films under different processing conditions. Free charge generation and recombination kinetics of (**i**) BM-Cl:BO-4Cl (CF), (**j**) BM-ClEH:BO-4Cl (CF), (**k**) BM-ClEH:BO-4Cl (THF) and (**l**) BM-HEH:BO-4Cl (THF) films under different post-treatment conditions.

## Ternary devices

To further improve the PCE of non-halogen solvent processed AMS-OSCs, ternary blending strategy was tried and SMD B1[49] was found to be an effective guest donor (Supplementary Fig. 34). ASM-OSCs based on THF-fabricated B1:BO-4Cl active layer delivered a PCE of 9.5% due to a low FF value (48.2%, Supplementary Table 16). B1 component with a small weight ratio of 10% was added into BM-ClEH:BO-4Cl (THF) to significantly enhance $J_{SC}$ and thereby enable THF-processed ASM-OSCs to achieve an optimal PCE of up to 16.1% (Supplementary Table 16)[60]. With further increase of B1 content in BM-ClEH:BO-4Cl, all parameters of devices will decrease. It is worth mentioning that 16.1% PCE represents one of the highest value for non-halogen solvent processed ASM-OSCs among currently reported results and is also comparative to the state-of-the-art ASM-OSCs devices fabricated by halogen-containing solvents[30–37].

Understanding the fundamental working mechanism and role of B1 in ternary matrix is significative for subsequent works. AFM and TAS measurements of ternary blends were carried out, shown in Supplementary Figs. 32, 33, respectively. It was observed that RMS values of BM-ClEH:BO-4Cl (THF) happen a sharp drop from 3.45 to 1.18 nm followed by a gradual decline when B1 was added with small weight ratios (5%, 10% and 20%). However, when B1 ratio increases to 40%, the surface of ternary film suddenly becomes rough (1.91 nm), possibly because B1 and BM-ClEH form a larger alloy due to similar molecular structure and appropriate proportion. Moreover, the $V_{OC}$ of ternary devices decrease along with the increase of B1 ratio, both indicating an alloy model working mechanism. TAS measurement shows that

optimal ternary blend displays the highest exciton quenching at 100 ps (Supplementary Fig. 33 and Fig. S34c), suggesting that the possibility of charge separation under normal operational device conditions is higher thereby capable of enhancing the $J_{SC}$ and EQE even B1 does not cause any substantial expansion of the blend absorption range.

## Devices stability tests

Device stability is another key criterion to evaluate the potential of OSC$_S$ for further applications[61–64]. Stability test of optimal ternary device based on BM-ClEH:B1:BO-4Cl (weight ratio of 0.9:0.1:1.2) was carried out under thermal degradation and maximum power point (MPP) tracking (Supplementary Figs. 34d and 35). Under a continuous TA at 80 °C in a N$_2$ glovebox without encapsulation, THF-processed devices are able to maintain over 60% initial PCE after 200 h, slightly better than that of CF-processed ones due to improved $J_{SC}$ along with burning time. MPP tracking stability of ternary devices largely outperform thermal stability that can maintain over 80% initial PCE ($T_{80}$) after 200 h and 70% initial PCE ($T_{70}$) after 450 h, demonstrating great potential of BM-ClEH:B1:BO-4Cl device for commercial application.

## Discussion

BM-ClEH was designed and synthesized to be a very efficient SMD for non-halogen solvent processed ASM-OSCs. Intramolecular Cl-S non-covalent interactions within BM-ClEH helps it to adopt a strong aggregation property through enhancing the rigidity of molecular backbone, which enables BM-ClEH achieve a good pre-aggregation state in THF and favorable blend morphologies for exciton

dissociation and carrier transport under TA treatment. Thus, THF-processed ASM-OSCs based on BM-ClEH:BO-4Cl delivered high PCEs of 15.0% in binary devices and 16.1% in ternary devices. Control SMD BM-HEH without Cl substitution exhibits weak aggregation, leading to inability of active layer to produce significant phase separation and free charge under THF processing, therefore barely functional. Overall, strong aggregation property of SMD is critical for realizing highly-efficient ASM-OSCs based on non-halogen solvent processing.

## Methods

### DSC measurements
Crystalline properties of three SMDs were characterized by differential scanning calorimetry (DSC) measurements on METTLER TOLEDO TGA/DGA3 + . The mass of sample is ~3.0 mg, and heating rate and cooling rate are both 5 °C/min.

### Temperature-variable NMR tests
The concentration of test sample is 5 mg/mL in CDCl$_3$. Range of test temperature for NMR (Bruker AVANCE 400 MHz) spectra is from 20 °C to 56 °C, lower than boiling point of deuterated reagent. $^1$H NMR signals were acquired every 4 °C.

### SCLC measurements
The structure of electron-only devices is ITO/ZnO/active layer/PFN-Br:MA/Ag and the structure of hole-only devices is ITO/PEDOT:PSS/active layers/MoOx/Ag. The fabrication conditions of active layer films are same with those for ASM-OSCs. The charge mobilities are generally described by the Mott-Gurney equation:

$$J = \frac{9}{8} \varepsilon_r \varepsilon_0 \mu \frac{V^2}{L^3} \tag{2}$$

where $J$ is the current density, $\varepsilon_0$ is the permittivity of free space (8.85 × 10$^{-14}$ F/cm), $\varepsilon_r$ is the dielectric constant of used materials, $\mu$ is the charge mobility, $V$ is the applied voltage and L is the active layer thickness. The $\varepsilon_r$ parameter is assumed to be 3, which is a typical value for organic materials.

### UPS measurements
Ultraviolet Photoelectron Spectroscopy (UPS) was performed by PHI 5000 VersaProbe III with He I source (21.22 eV) under an applied negative bias of 10.0 V.

### ASM-OSCs fabrication and characterization
ASM-OSCs were fabricated by using a conventional device structure of ITO/PEDOT:PSS/active layers/PFN-Br:MA/Ag. The patterned ITO-coated glass substrates were first scrubbed by detergent and then cleaned inside an ultrasonic bath by using deionized water, acetone and isopropanol subsequently, and dried overnight in an oven. The glass substrates were treated by UV-Ozone for 30 min before use to improve its work function and clearance. PEDOT:PSS (Al4083 from Hareus) was spin-casted onto the ITO substrates at 7500 rpm for 30 s, and then dried at 160 °C for 15 min in N$_2$ atmosphere. The fully dissolved blend solution of BM-Cl:BO-4Cl in CF, BM-ClEH:BO-4Cl in CF, BM-ClEH:BO-4Cl in THF and BM-HEH:BO-4Cl in THF (weight ratio of 1:1 and total concentration of 20 mg/mL) was spin-casted at 2000 rpm for 30 s onto PEDOT:PSS film followed by a thermal annealing (80–150 °C) or solvent evaporation annealing (chlorobenzene, 60–100 s). A thin PFN-Br:MA layer (0.5 mg/mL in methanol and 0.25% wt% melamine, 3000 rpm) was coated on the active layer, followed by the deposition of Ag (evaporated under 3 × 10$^{-4}$ Pa through a shadow mask). The fabrication process of ternary devices is same as the binary devices by replacing the blend solution with BM-ClEH:B1:BO-4Cl (donor:acceptor weight ratio of 1:1 with B1 ratio of 0, 5, 10, 20, 40 and 100 wt% in donor, total concentration of 20 mg/mL). The optimal active layer thickness

measured by a Bruker Dektak XT stylus profilometer was about 110 nm. The current density-voltage (J-V) curves of devices were measured using a Keysight B2901A Source Meter in glove box under AM 1.5 G (100 mW cm$^{-2}$) using an Enlitech solar simulator. The device contact area was 0.042 cm$^2$, device illuminated area during testing was 0.0324 cm$^2$, which was determined by a mask. The EQE spectra were measured using a Solar Cell Spectral Response Measurement System QE-R3011 (Enlitech Co., Ltd.). The light intensity at each wavelength was calibrated using a standard monocrystalline Si photovoltaic cell. The MPP tracking was carried out upon Epoxy encapsulated devices under 1-sun white LED array in air. The whole tracking condition (temperature and humidity) is co-controlled by air-conditioner and blowing cooling setups.

### TAS measurements
TAS was measured with an amplified Ti:sapphire femtosecond laser (800 nm wavelength, 50 fs, 1 kHz repetition; Coherent Libra) and a Helios pump/probe setup (Ultrafast Systems). The 400 nm pump pulses with a pump fluence of 0.5 or < 0.3 μJ/cm$^2$ were obtained by frequency doubling the 800 nm fundamental regenerative amplifier output. The white-light continuum probe pulses were generated by focusing a small portion of the regenerative amplifier's fundamental 800 nm laser pulses into a 2 mm sapphire crystal.

### Reporting summary
Further information on research design is available in the Nature Portfolio Reporting Summary linked to this article.

## Data availability
The data that support the findings of this study are presented in Supplementary Information and Source Data file. The source data for Fig. 4b, Table 1, Supplementary Fig. 16a–c and Supplementary Fig. 34a generated in this study are provided as a Source Data file. Source data are provided with this paper.

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

## Acknowledgements

G. Li acknowledges the support from Research Grants Council of Hong Kong (Project Nos 15221320, 15307922, C5037-18G, C4005-22Y), RGC Senior Research Fellowship Scheme (SRFS2223-5S01), the Hong Kong Polytechnic University: Sir Sze-yuen Chung Endowed Professorship Fund (8-8480), RISE (Q-CDBK), PRI (Q-CD7X), G-SAC5 and Guangdong-Hong Kong-Macao Joint Laboratory for Photonic-Thermal-Electrical Energy Materials and Devices (GDSTC No. 2019B121205001). W.G. thanks the support from Scientific Research Funds of Huaqiao University (605-50Y23024). R.M. thanks the PolyU Distinguished Postdoctoral Fellowship (1-YW4C). H.L. thanks the support of Sichuan Science and Technology Program (2023NSFSC0990). Shanghai Beamline BL02U2 and BL16B1 are appreciated for GIWAXS and GISAXS data acquisition, respectively. The authors gratefully acknowledge the cooperation of the beamline scientists at BSRF-1W1A beamline and help in UPS tests from Nanjing Demo Science Limited.

## Author contributions

W.G., R.M., and G.L. conceived the ideas. W.G. designed and synthesized the three small molecule donors, carried out the DSC, temperature-variable UV-vis absorption of solution, temperature-variable NMR measurements. R.M. performed the device characterization and optimization, conducted the J-V, EQE, Jph-Veff, SCLC, temperature-variable UV-vis absorption of films measurements and fabricated the samples for morphology test. R.M., T.A.D.P., M.L., J.W. and G.L. performed the TAS measurement and analyzed the data. H.L., C.Y. and P.C. conducted the AFM, TEM, GIWAXS and GISAXS measurements. C.Z. conducted the DFT calculations. W.G., R.M., Z.W., A.K.-Y.J. and G.L. contributed to the preparation of this manuscript. All authors have commented on this manuscript.

## Competing interests

The authors declare no competing interests.
