## [Peer Review File · Nature Communications]

Efficient all-small-molecule organic solar cells processed with non-halogen solventREVIEWER COMMENTS

Reviewer #1 (Remarks to the Author):

In this work, Gao et al. reported a small molecule donor, namely BM-CIEH, that shows a strong H-aggregation property due to intramolecular Cl-S non-covalent interaction. With THF as the processed solvent, BM-CIEH-based devices achieve high PCEs of 15.0% for binary blends and 16.1% for ternary blends, which are among the highest values for such kinds of devices as far as I know. By comparing with a J-aggregated small molecule donor BM-HEH without Cl-substitution, the authors demonstrated the complex influence of H-aggregation property on the pre-aggregation state of donor molecule in solution, morphology of active layer processed under different conditions and charge generation, and thus the importance of H-aggregation property for realizing highly-efficient ASM-OSCs under green-solvent processing. Moreover, the optimal device can be achieved by thermal annealing treatment and can maintain relatively good stability under MPP tracking conditions. This manuscript is well-organized and written. In view of the above, I recommend this manuscript can be accepted by Nature Communications after some revision:

1. In this manuscript, the authors did not provide the absorption spectra of blend films. Actually, changes in absorption spectra for blend films with temperature can be used to observe the aggregation state of donor and acceptor components. Moreover, they can be also used to calculate DMT values.
2. Under a similar optimized method, the performance of the BM-CIEH-based device is higher in THF than that in CF, which is quite uncommon. How about the solubility of BM-CIEH in CF and what is the effect of different solubility on the film morphology? How about the device performance with CS₂ as the processed solvent?
3. As the authors mentioned, strong H-aggregation property is conducive to improving molecular pre-aggregation in and corresponding film micromorphology, which is induced by intramolecular Cl-S non-covalent interaction. How to prove the existence of intramolecular Cl-S non-covalent interaction in BM-CIEH and BM-Cl.
4. I suggest the authors measure the photoluminescence (PL) spectra of neat and blend films to more clearly demonstrate the change of molecular packing before and after TA treatment, and charge transfer between donor and acceptor because packing enhancement of pure films usually leads to the enhancement of their PL, and conversely for blend films.
5. With 10 wt% B1 addition, the ternary devices achieve an optimal PCE of up to 16.1% with significantly enhance JSC, while the B1-based device shows very poor efficiency and JSC. What is the fundamental working mechanism in the ternary matrix? More discussion about that should be added in the revised manuscript.
6. Have the authors tried to use L8-BO and BTP-eC9 as acceptor materials for fabricating high-performance ASM-OSCs devices, instead of BO-4Cl to fabricate devices by using THF?

Reviewer #2 (Remarks to the Author):

In this manuscript entitled 'Strongly H-aggregated small molecule donor for non-aromatic green solvent processed highly-efficient and stable all-small-molecule organic solar cells' the authors achieve record power conversion efficiencies (15.0% in binary and ternary 16.1% in ternary) for green solvent processed all-small-molecule organic solar cells. By comparing three small molecule donors, the authors emphasize that strong H-aggregation behavior can maintain pre-aggregation state in THF, leading to good film-forming morphology and finally reflecting on high device performance. The effort from co-authors is appreciated in boosting the performance of the small molecule solar cells. However, there are severe novelty and urgency concerns raised when I looked at the manuscript. Some of the comments to improve the manuscript for further submissions can be found below:

1. The selection of THF as a green solvent should be re-considered, what is the motivation behind using THF as green solvent, what justifies that it is a novel green solvent? Capello Christian et al. reported a thorough assessment of green solvent (Green Chem. 2007, 9, 927-934. <https://doi.org/10.1039/B617536H>). THF is not recommendable from an environmental perspective: it causes high environmental impacts during petrochemical production; it has a complex petrochemical production route requiring several production steps. However, on line 107, the authors claim THF is cheap and environmental friendly, and do not depend on petroleum products without any citation.
2. The authors claim in the abstract 'green solvent processed ASM-OSCs are rarely reported and their power conversion efficiencies (PCEs) are difficult to exceed 10%.' Recently, THF-processed binary ASM-OSCs reaching 14% PCE have been reported (Adv. Funct. Mater. 2023, 2300778. <https://doi.org/10.1002/adfm.202300778>). However, this paper is not cited in the introduction section or Figure 4d. There are further work on green solvents (not particularly on small molecules yet significant breakthroughs in the field) yet the manuscript does not even touch base.
3. The energy levels of BM-Cl, BM-CIEH, and BM-HEH are missing. Regarding to the extremely low PCE of BM-HEH (<1%), it is important to show that such low PCE is not from the energy level mismatch by replacing the substitution group from -Cl to -H. Precise energy level determination is very important using photoelectron spectroscopy.
4. The link between the device FF and the hole/electron mobility ratio is confusing. From Figure 4f, CM-CIEH (CF) has an imbalance mobility ratio of over 20, which can be deduced as strong bi-molecular recombination, however, the FF of the device is still high (69.8%). Furthermore, we can see that by changing the donors and solvent, the most obvious mobility change is electrons rather than holes. The reasons that may cause such acceptor mobility change should be further elaborated. Additionally, the caption of Figure 4f should be more precise and the authors should discuss 4f in the main text as well.
5. From Table 1, all as-cast devices have low PCE (<5%) no matter using CF or THF. All the high-performance devices are based on film thermal annealing treatment. Therefore, the link between the pre-aggregate state in solution and final device performance is not convincing and needs further discussion.
6. In Figure 8d, the reason for testing the stability of the ternary device is lacking. A comparison between the green solvent processed device and the normal CF processed device will be useful.

Reviewer #3 (Remarks to the Author):

Gao et al report the “pre-aggregation” effects on the device performances fabricated by THF, and the ternary solar cells lead to an efficiency exceeding 16%. Relating the molecular structure with morphology is quite important. However, honestly speaking, the morphology for the ASM-OSCs is quite complicated. Because it is not only influenced by the solvent effect in solution, but also other many important factors such as Flory-Huggin parameter, crystalline, thermal effect, and solvent effect in film formation and evaporation. Hence, the whole paper doesn't persuade me from the following two main aspects: a) What is the relation-ship between molecular structure with H-aggregates? Whether there are really H aggregates in your solution? b) Pre-aggregation for polymer is quite important, due to its limited diffusion speed during film formation. however, for small molecules, from my opinion, neglecting the miscibility and film formation process to talk about morphology is not reasonable. Therefore, I don't recommend this paper to publish in Nature Communications, and my detail questions are as follows:

1. In the absorption spectrum especially in solution, why it could use I(0-0) and I (0-1) to analyze the H and J aggregates? Which peak is attributed to internal charge transfer? In the film, how do you distinguish the differences between H/J aggregates and ICT absorption /aggregation absorption band? In fact, for small molecules, I am not convinced that it adopts H or J aggregates in the solution, although it definitely exists solvation effect. The absorption spectrum variation with temperature in the diluted solution is maybe only because of rotation, and has nothing to do with H/J aggregates.
2. Before the part “Morphology, mobility and overall interaction for neat films” the author demonstrated that Cl-S non-covalent interaction within BM-Cl and BM-CLEH enhances molecular rigidity as a result to obtain strong H-aggregation property. My question is what is the relation-ship between molecular rigidity and H-aggregation property? Additionally, whose intermolecular force is larger, in your opinion, H or J?
3. The H shifts with temperature in NMR is used to explain the molecular rigidity. What is your concentration for NMR (10 mg/ml)? What is your concentration for UV-vis (10-5 mg/ml) ? If there is a pre-aggregation in UV-vis absorption, and there would be strong pre-aggregation in your NMR test. It is puzzled me that the variation of H shifts with temperature is only related to molecular rotation and the variation of absorption spectrum with temperature is only related to pre-aggregation in the analysis.
4. In BM-HEH:BO4Cl, its poor efficiency may be due to its unsatisfactory hetero-intermolecular interaction, and it may not be related to the "pre-aggregation" you mentioned. Both the halogen substituent and the alkyl chain modification have a great influence on the surface energy.
5. For the BP (boiling point) for THF is higher than CF, its film is always more condensed than that spin-coated by CF, which shouldn't be contributed this only to the solvent-effect in solution. Besides, as for BO-4Cl (THF) film undergoes largest thermal motion deduced from MDT, however, its electron mobility is only twice times higher, which is much inferior compared to others (one order magnitude improvement after TA, Figure 3). So, in your opinion, what is the key point to induce thermal packing?
6. The TAS part is not well organized, although the absorption of intermediate states covering 700-800 nm seems interesting, however, the explanation is not convincing. From time-scale, it emerges even faster than the GSB signal of donor, how does this happen? Further, there are many small errors in this

part. Such as “GBS”, “stark effect then EA features” “overlap with the EA on the said range” and so on. In Figure S23, what are the molecules of DG7, DG8? Figure S22, what is the wavelength of singlet exciton? In stability measurements, what is the condition? N2 atmosphere or exposure to the air after encapsulation?

Point-to-point Response to Referees Letter

Reviewers' Comments to Authors:

Reviewer #1:

In this work, Gao et al. reported a small molecule donor, namely BM-CIEH, that shows a strong H-aggregation property due to intramolecular Cl-S non-covalent interaction. With THF as the processed solvent, BM-CIEH-based devices achieve high PCEs of 15.0% for binary blends and 16.1% for ternary blends, which are among the highest values for such kinds of devices as far as I know. By comparing with a J-aggregated small molecule donor BM-HEH without Cl-substitution, the authors demonstrated the complex influence of H-aggregation property on the pre-aggregation state of donor molecule in solution, morphology of active layer processed under different conditions and charge generation, and thus the importance of H-aggregation property for realizing highly-efficient ASM-OSCs under green-solvent processing. Moreover, the optimal device can be achieved by thermal annealing treatment and can maintain relatively good stability under MPP tracking conditions. This manuscript is well-organized and written. In view of the above, I recommend this manuscript can be accepted by Nature Communications after some revision:

1. In this manuscript, the authors did not provide the absorption spectra of blend films. Actually, changes in absorption spectra for blend films with temperature can be used to observe the aggregation state of donor and acceptor components. Moreover, they can be also used to calculate DM_T values.

Reply: We thank the reviewer for the good suggestion. The absorption spectra of BM-Cl:BO-4Cl (CF), BM-CIEH:BO-4Cl (CF), BM-CIEH:BO-4Cl (THF) and BM-HEH:BO-4Cl (THF) blend films under different TA temperature and SVA time have been measured, which are shown in Supplementary Fig. 10. The DM_T values of four blend films under different TA temperature were also calculated and compared (Supplementary Fig. 11). Moreover, by combining absorption spectra under different TA temperature and DM_T values, we discussed in detail the change of aggregation state of donor and acceptor components in the blend films. The corresponding discussion has been added into the revised manuscript as following:

The corresponding revision in Manuscript (yellow highlight) is listed below:

When coming into blend films, molecular aggregation behaviors and thermal motion will become more complex. The absorption spectra of blend films under different TA temperature and SVA time were measured (Supplementary Fig. 10). It can be clearly observed that the large addition of BO-4Cl will partially destroy H-aggregation of SMDs, especially in BM-CIEH. However, as TA temperature increases, the 0-0 characteristic peak gradually appears and intensifies in BM-Cl:BO-4Cl and BM-CIEH:BO-4Cl films, which can be probably attributed to amorphous SMD components

are inability to fully revert to *H*-aggregates but leads to formation of *J*-aggregates under TA action. It should be noted that the stronger *0-0* and *0-1* peaks of BM-Cl:BO-4Cl are derived from a stronger crystallinity of BM-Cl than BM-CIEH though a faster rate of thermal motion of BM-CIEH in CF-casted film. BM-CIEH:BO-4Cl (THF) seems to be more sensitive to TA treatment and achieved stronger *0-0* and *0-1* peaks than that of BM-CIEH:BO-4Cl (CF) due to a more significant molecule motion by thermal of BO-4Cl in THF-casted film than in CF-casted film, which determines a larger DM_T values of BM-CIEH:BO-4Cl (THF) than BM-CIEH:BO-4Cl (CF) (Supplementary Fig. 11). Likewise, the smaller DM_T value of BM-Cl:BO-4Cl (CF) than that of BM-CIEH:BO-4Cl (CF) can be due to weaker thermal motion of BM-Cl. It seems that the rapid thermal motion of BO-4Cl in turn accelerates the motion of BM-CIEH to facilitate better *H*- and *J*-aggregates in THF-casted blend films. SVA is able to give molecules more free movement space in blend films and quickly make donor and acceptor reach a good aggregation state. Such large freedom of molecule motion is beneficial for BM-Cl or BM-CIEH to form *H*-aggregates compared to TA-treated blend films. However, for BM-HEH, neither TA nor SVA can recover its aggregation state as same as in pure film, indicating that it is completely an amorphous state in the blend films.

Fig. S10. Normalized UV-vis absorption spectra of blend films under different TA temperature and SVA time: (a,e) BM-Cl:BO-4Cl (CF), (b,f) BM-CIEH:BO-4Cl (CF), (c,g) BM-CIEH:BO-4Cl (THF), (d,h) BM-HEH:BO-4Cl (THF).

Fig. S11. Deviation metric of redshifts measured from temperature-variable UV-vis absorption spectra vs temperature for blend films.

2. Under a similar optimized method, the performance of the BM-CIEH-based device is higher in THF than that in CF, which is quite uncommon. How about the solubility of BM-CIEH in CF and what is the effect of different solubility on the film morphology? How about the device performance with CS₂ as the processed solvent?

Reply: Thanks for the reviewer's comments. It can be found in Table S7 and S8 that the PCE of ASM-OSCs based on BM-CIEH:BO-4Cl processed with THF is lower than that processed with CF under same SVA time and both can achieve optimal PCEs under SVA 80 s (Fig. R1a). However, under TA condition, the PCEs of ASM-OSCs based on BM-CIEH:BO-4Cl (THF) is indeed higher than that based on BM-CIEH:BO-4Cl (CF) at 80°C and 100°C, but lower at 120°C. The optimal TA temperature for best ASM-OSCs based on BM-CIEH:BO-4Cl (CF) (PCE of 13.6%) and BM-CIEH:BO-4Cl (THF) (PCE of 15.0%) is 120°C and 100°C, respectively (Fig. R1b). By calculating DM_T values and ratio of DM_T vs temperature of BO-4Cl in neat film, we found that thermal motion of BO-4Cl molecule in THF-casted film is faster than that in CF-cast film (Fig. 3i and 3j). Thus, BM-CIEH:BO-4Cl (THF) blend film is able to achieve better molecular packing (or morphology) at relatively lower TA temperature compared to BM-CIEH:BO-4Cl (CF) blend film, which can be also supported by absorption spectra of two blend films under different TA temperature (Supplementary Fig. 10). Therefore, at relatively low (80°C and 100°C) TA temperature, it is reasonable that the PCE of ASM-OSCs based on BM-CIEH:BO-4Cl processed with THF is higher than that processed with CF. Moreover, under optimal TA temperature (100°C for BM-CIEH:BO-4Cl (THF) and 120°C for BM-CIEH:BO-4Cl (CF)), BM-CIEH:BO-4Cl (THF) active layer can achieve higher hole mobility (more balance charge transport, Table S13) and more order π - π stacking (Table S14). Therefore, it is reasonable that BM-CIEH:BO-4Cl-based ASM-OSCs processed with THF can achieved higher PCE than that processed with CF under a similar optimized method.

Fig. R1. PCEs of ASM-OSCs based on BM-CIEH:BO-4Cl processed with CF or THF under (a) SVA and (b) TA.

The solubility of BM-CIEH in CF were tested by using method mentioned in SI, and was estimated to be between 50 and 66 mg/mL, which is greater than that in THF. In a certain range, the better the solubility of SMDs may be the better the miscibility with acceptor, which may be conducive to form better phase separation and domain size. If the solubility of SMD is too good, it may affect crystallization property of SMD itself, resulting in poor morphology. However, it is also difficult to determine the effect of SMD's solubility on active layer morphology because solvent's difference will have influence on the formation of film thus morphology.

We have fabricated the ASM-OSCs based on BM-CIEH:BO-4Cl with CS₂ as processing solvent. The weight ratio of donor and acceptor is 1:1, and total concentration of blend solution is 18 mg/mL. The spin-coating speed is 3500 rpm. The optimal device was obtained under condition of TA at 120°C, which achieved a PCE of 10.9%. We have added this result into revised manuscript as following:

The corresponding revision in Manuscript (yellow highlight) is listed below:

To further demonstrate the potential of BM-CIEH in different green solvents and compatibility with other acceptor materials, ASM-OSCs with carbon disulfide (CS₂) as processing solvent or BTP-eC9 and L8-BO as electron acceptors were fabricated. Optimized J-V curves of corresponding devices were displayed in Supplementary Fig. 18. with photovoltaic parameters summarized in Supplementary Table 11. As a result, ASM-OSCs based on BM-CIEH:BO-4Cl processed with CS₂ can achieve close to 11% PCE along with a 71.2% FF under TA post-treatment at 120°C, which is a relatively high efficiency for such low boiling point (46.2 °C) solvent processing and a rare case for OSCs processed with CS₂. When altering acceptor materials, ASM-OSCs based on BM-CIEH:BTP-eC9 and BM-CIEH:L8-BO fabricated with THF gained 12.6% and 13.1% PCE, respectively, indicating a good morphology compatibility of BM-CIEH with different acceptors in green solvent processing.

Fig. S18. J - V curves for ASM-OSCs based on (a) BM-CIEH:BO-4Cl fabricated with CS_2 , (b) BM-CIEH:BTP-eC9 (THF) and (c) BM-CIEH:L8-BO (THF). EQE spectra for ASM-OSCs based on (d) BM-CIEH:BO-4Cl fabricated with CS_2 , (e) BM-CIEH:BTP-eC9 (THF) and (f) BM-CIEH:L8-BO (THF).

Table S11 Photovoltaic parameters for ASM-OSCs based on BM-CIEH:BO-4Cl fabricated with CS_2 , BM-CIEH:BTP-eC9 (THF) and BM-CIEH:L8-BO (THF).

Active Layer	V_{oc} (V)	J_{sc} (mA cm^{-2})	J_{sc}^a (mA cm^{-2})	FF (%)	PCE (%)
BM-CIEH:BO-4Cl (CS_2 , 100°C)	0.865	14.78	13.98	66.5	8.5
BM-CIEH:BO-4Cl (CS_2 , 120°C)	0.850	17.95	17.29	71.2	10.9
BM-CIEH:BTP-eC9 (THF, 100°C)	0.866	20.50	19.71	70.7	12.6
BM-CIEH:BTP-eC9 (THF, 120°C)	0.853	19.72	18.95	72.6	12.2
BM-CIEH:L8-BO (THF, 100°C)	0.908	21.27	20.65	60.3	11.7
BM-CIEH:L8-BO (THF, 120°C)	0.894	21.75	21.02	67.1	13.1

^aIntegrated from EQE spectra.

3. As the authors mentioned, strong H-aggregation property is conducive to improving molecular pre-aggregation in and corresponding film micromorphology, which is induced by intramolecular Cl-S non-covalent interaction. How to prove the existence of intramolecular Cl-S non-covalent interaction in BM-CIEH and BM-Cl.

Reply: We thank the reviewer for your insightful comment. Generally, two necessary conditions are required for the formation of intramolecular non-covalent interaction: (i) an atom has an empty orbital and another atom has a lone pair of electrons; (ii) the space distance of two atoms is less than the sum of van der Waals radii of two atoms. Some previous literatures have discussed the existence of intramolecular Cl-S non-covalent interaction, such as *Chem. Rev.* 2017, 117, 10291-10318 by H. Huang and T. J. Marks et al. and *Adv. Sci.* 2020, 7, 2000509 by F. He et al. The electron configuration of S atom is $1s(2)2s(2)2p(6)3s(2)3p(4)$ while the electron configuration of Cl atom is

$1s(2)2s(2)2p(6)3s(2)3p(5)3d(0)$. S atom has a lone pair of electrons and Cl atom has empty 3d orbitals. In order to prove the existence of intramolecular Cl-S non-covalent interaction in BM-CIEH (or in BM-Cl), we carried out density functional theory (DFT) calculations where long alkyl chain were shorted to be ethyl. The optimal molecular conformation of BM-CIEH (or BM-Cl) was shown in Supplementary Fig. 2. The space distance between S and Cl atoms on adjacent two thiophenes within BM-CIEH (BM-Cl) is found to be 304 pm, which is smaller than the sum (361 pm) of van der Waals radii of S (185 pm) and Cl (176 pm) atoms. Therefore, intramolecular Cl-S non-covalent interaction can be considered to exist in BM-CIEH and BM-Cl. Moreover, by mapping the intramolecular weak interaction, we further find that the non-covalent interaction of Cl-S is stronger than that of H-S (marked by blue circle). We have added DFT calculations into revised manuscript as following:

The corresponding revision in Manuscript (yellow highlight) is listed below:

Density functional theory (DFT) calculations were performed to understand effects of Cl-substitution on SMDs on a microcosmic aspect where long alkyl chains were shortened to be ethyl (Supplementary Fig. 2). In optimal molecular configuration, the space distance of Cl and S atoms on adjacent two thiophenes within BM-CIEH (BM-Cl) is found to be 304 pm, significantly smaller than the sum (361 pm) of van der Waals radii of S (185 pm) and Cl (176 pm) atoms, indicating the existence of intramolecular Cl-S non-covalent interaction.⁵² Moreover, by further mapping intramolecular weak interactions within BM-CIEH (BM-Cl), we observed that Cl-S interaction is remarkably larger than that of H-S, revealing Cl-substitution is favorable to endow stronger intramolecular non-covalent interaction. With this effect, BM-CIEH obtains a slightly flatter molecular skeleton and a smaller binding energy of bimolecular that adopted a *H*-aggregation mode (-5.53 eV for BM-CIEH and -5.15 eV for BM-HEH) than these of BM-HEH, which suggests that Cl-substitution is conducive to realize better intermolecular aggregation/packing.

Fig. S2. DFT calculation results for BM-CIEH and BM-HEH monomer and dimer.

4. I suggest the authors measure the photoluminescence (PL) spectra of neat and blend films to more clearly demonstrate the change of molecular packing before and after TA

treatment, and charge transfer between donor and acceptor because packing enhancement of pure films usually leads to the enhancement of their PL, and conversely for blend films.

Reply: *Thanks for the reviewer's good suggestions. We have measured the PL spectra of BM-Cl (CF), BM-CIEH (CF), BM-CIEH (THF), BM-HEH (THF), BO-4Cl (CF), BO-4Cl (THF) neat films and BM-Cl:BO-4Cl (CF), BM-CIEH:BO-4Cl (CF), BM-CIEH:BO-4Cl (THF), BM-HEH:BO-4Cl (THF) blend films without/with TA and SVA treatments. PL spectra show that TA and SVA improve molecular aggregation and/or ordering that leads to emission of neat films red-shift or enhancement while quenching efficiencies of blend films are improved. We have added the PL measurements into revised manuscript as following:*

The corresponding revision in Manuscript (yellow highlight) is listed below:

Photoluminescence spectra (PL) of BM-Cl (CF), BM-CIEH (CF), BM-CIEH (THF), BM-HEH (THF), BO-4Cl (CF) and BO-4Cl (THF) neat films without/with TA or SVA were measured. As shown in Supplementary Fig. 12, TA and SVA treatments are able to improve molecular aggregation and/or ordering to induce PL emission enhancements (for SMDs) or red-shifts (for BO-4Cl). On the whole, TA effect is more obvious than SVA which is beneficial to achieve better film morphology. Bandgaps (E_g s) of investigated materials are determined by calculating the intersection of absorption and PL spectra (Supplementary Fig. 13), which are 1.831, 1.866, 1.857, 1.851, 1.439 and 1.431 eV, respectively. In addition, ultraviolet photoelectron spectroscopy (UPS) measurements were performed in order to accurately probe energy levels of SMDs and SMA (Supplementary Fig. 14). The Fermi levels (E_F) and valence bands (VB) of above films were fitted to be -4.34/0.55, -4.56/0.49, -4.50/0.56, -4.52/0.50, -4.62/0.98 and -4.45/1.12 eV, respectively, corresponding to highest occupied molecular orbital energy levels (E_{HOMO}) of -4.98, -5.05, -5.06, -5.02, -5.60, -5.57 eV, respectively. The lowest unoccupied molecular orbital energy levels (E_{LUMO}) were also obtained as -3.06, -3.18, -3.20, -3.17, -4.16 and -4.14 eV, respectively, according $E_{LUMO} = E_{HOMO} + E_g$. Sufficient energy levels offsets between E_{HOMOS} or E_{LUMOS} of SMDs and SMA can effectively ensure exciton transfer. Interestingly, the Fermi level of THF-processed films elevates relative to CF-processed one and VB widens, especially for BO-4Cl, which is likely to facilitate hole transfer and transport. PL spectra of BM-Cl:BO-4Cl (CF), BM-CIEH:BO-4Cl (CF), BM-CIEH:BO-4Cl (THF) and BM-HEH:BO-4Cl (THF) blend films exhibited significant quenching with efficiencies reach more than 90% after TA and SVA, indicating good charge transfer from SMA to SMDs (Supplementary Fig. 15).

Fig. S12. PL spectra of (a) BM-Cl (CF), (b) BM-CIEH (CF), (c) BO-4Cl (CF), (d) BM-CIEH (THF), (e) BM-HEH (THF) and (f) BO-4Cl (THF) neat films without/with TA or SVA treatments.

Fig. S13. Intersections of absorption and PL spectra for (a) BM-Cl (CF), (b) BM-CIEH (CF), (c) BM-CIEH (THF), (d) BM-HEH (THF), (e) BO-4Cl (CF) and (f) BO-4Cl (THF) neat films.

Fig. S15. PL quenching for (a) BM-Cl:BO-4Cl (CF), (b) BM-CIEH:BO-4Cl (CF), (c) BM-CIEH:BO-4Cl (THF) and (d) BM-HEH:BO-4Cl (THF) blend films.

5. With 10 wt% B1 addition, the ternary devices achieve an optimal PCE of up to 16.1% with significantly enhance J_{SC} , while the B1-based device shows very poor efficiency and J_{SC} . What is the fundamental working mechanism in the ternary matrix? More discussion about that should be added in the revised manuscript.

Reply: Thanks for the reviewer's insightful comment. Although PCE of ASM-OSCs based on B1:BO-4Cl (THF) is not high, it is found to be a suitable third component adding to BM-CIEH:BO-4Cl (THF) system, resulting in significantly improved J_{SC} and thus PCE. It is significative to understand the fundamental working mechanism and role of B1 in this ternary matrix for subsequent works. Therefore, AFM and TAS measurements were carried out, which are shown in Supplementary Fig. 32 and Fig. 33, respectively. It was observed RMS value of B1:BO-4Cl (THF) (0.85 nm) is significantly lower than that of BM-CIEH:BO-4Cl (THF) (3.52 nm). When B1 was added into BM-CIEH:BO-4Cl (THF) with small ratio (5% wt, 10% wt and 20% wt), the RMS values of BM-CIEH:B1:BO-4Cl (THF) happen a sharp drop followed by a gradual decline. However, when the ratio of B1 increases to 40%, the surface of ternary film suddenly becomes rough, possibly because B1 and BM-CIEH can form a larger alloy due to similar molecular structure and appropriate proportions. Moreover, the Vocs of ternary ASM-OSCs decrease along with the increase of B1 ratio, therefore, the fundamental working mechanism of this ternary OSCs is a alloy model. TAS

measurements shows that optimal ternary blend displays the highest exciton quenching at 100 ps (Fig. 8c), suggesting that the possibility of charge separation under normal operational device conditions is higher thereby capable of enhancing the J_{sc} and EQE even B1 does not cause any substantial expansion of the blend absorption range. We have added discussion about that into revised manuscript:

The corresponding revision in Manuscript (yellow highlight) is listed below:

Understanding the fundamental working mechanism and role of B1 in ternary matrix is significant for subsequent works. AFM and TAS measurements of ternary blends were carried out, shown in Supplementary Fig. 32-33, respectively. It was observed that RMS values of BM-CIEH:BO-4Cl (THF) happen a sharp drop from 3.45 to 1.18 nm followed by a gradual decline when B1 was added with small weight ratios (5%, 10% and 20%). However, when B1 ratio increases to 40%, the surface of ternary film suddenly becomes rough (1.91 nm), possibly because B1 and BM-CIEH form a larger alloy due to similar molecular structure and appropriate proportion. Moreover, the V_{oc} s of ternary devices decrease along with the increase of B1 ratio, both indicating an alloy model working mechanism. TAS measurement shows that optimal ternary blend displays the highest exciton quenching at 100 ps (Supplementary Fig. 33 and Fig. 8c), suggesting that the possibility of charge separation under normal operational device conditions is higher thereby capable of enhancing the J_{sc} and EQE even B1 does not cause any substantial expansion of the blend absorption range.

Fig. S32. AFM height images of BM-CIEH:B1:BO-4Cl blend films with (a) 5%, (b) 10%, (c) 20%, (d) 40% and (e) 100% B1 weight ratios. Corresponding AFM phase images (f-j).

Fig. S33. 2D plots of TAS measurements for BM-CIEH:B1:BO-4Cl (THF) blend films with (a) 0%, (b) 10% and (c) 100% B1 weight ratio. Spectral cuts of TAS measurements after different time for BM-CIEH:B1:BO-4Cl (THF) blend films (d) 0%, (e) 10% and (f) 100% B1 weight ratio.

Fig. 8 | Green solvent processed ternary ASM-OSCs. (a) J - V curves of THF-processed ternary AMS-OSCs based on BM-CIEH:B1:BO-4Cl with different B1 weight ratio. (b) Corresponding EQE spectra. (c) Free charge generation and recombination kinetics of BM-CIEH:B1:BO-4Cl devices with 0%, 10% and 100% B1 addition. (d) Stability of optimal device under MPP tracking.

6. Have the authors tried to use L8-BO and BTP-eC9 as acceptor materials for fabricating high-performance ASM-OSCs devices, instead of BO-4Cl to fabricate devices by using THF?

Reply: Thanks for the reviewer's good suggestion. We have tried to use L8-BO and BTP-eC9 as acceptor materials to fabricate ASM-OSCs based on THF instead of BO-4Cl. Optimal PCEs of THF-processed ASM-OSCs based on BM-CIEH:BTP-eC9 (TA at 100°C) and BM-CIEH:L8-BO (TA at 120°C) are 12.6% and 13.1%, respectively, unfortunately, which are both lower than that of THF-processed ASM-OSCs based on BM-CIEH:BO-4Cl due to decreased J_{sc} or FF. We have added this result into revised manuscript as following:

The corresponding revision in Manuscript (yellow highlight) is listed below:

To further demonstrate the potential of BM-CIEH in different green solvents and compatibility with other acceptor materials, ASM-OSCs with carbon disulfide (CS₂) as processing solvent or BTP-eC9 and L8-BO as electron acceptors were fabricated. Optimized J - V curves of corresponding devices were displayed in Supplementary Fig. 18. with photovoltaic parameters summarized in Supplementary Table 11. As a result, ASM-OSCs based on BM-CIEH:BO-4Cl processed with CS₂ can achieve close to 11% PCE along with a 71.2% FF under TA post-treatment at 120°C, which is a relatively high efficiency for such low boiling point (46.2 °C) solvent processing and a rare case for OSCs processed with CS₂. When altering acceptor materials, ASM-OSCs based on BM-CIEH:BTP-eC9 and BM-CIEH:L8-BO fabricated with THF gained 12.6% and 13.1% PCE, respectively, indicating a good morphology compatibility of BM-CIEH with different acceptors in green solvent processing.

Fig. S18. J - V curves for ASM-OSCs based on (a) BM-CIEH:BO-4Cl fabricated with CS₂, (b) BM-CIEH:BTP-eC9 (THF) and (c) BM-CIEH:L8-BO (THF). EQE spectra for

ASM-OSCs based on (d) BM-CIEH:BO-4Cl fabricated with CS₂, (e) BM-CIEH:BTP-eC9 (THF) and (f) BM-CIEH:L8-BO (THF).

Table S11 Photovoltaic parameters for ASM-OSCs based on BM-CIEH:BO-4Cl fabricated with CS₂, BM-CIEH:BTP-eC9 (THF) and BM-CIEH:L8-BO (THF).

Active Layer	V_{oc} (V)	J_{sc} (mA cm ⁻²)	$J_{sc}^{a)}$ (mA cm ⁻²)	FF (%)	PCE (%)
BM-CIEH:BO-4Cl (CS ₂ , 100°C)	0.865	14.78	13.98	66.5	8.5
BM-CIEH:BO-4Cl (CS ₂ , 120°C)	0.850	17.95	17.29	71.2	10.9
BM-CIEH:BTP-eC9 (THF, 100°C)	0.866	20.50	19.71	70.7	12.6
BM-CIEH:BTP-eC9 (THF, 120°C)	0.853	19.72	18.95	72.6	12.2
BM-CIEH:L8-BO (THF, 100°C)	0.908	21.27	20.65	60.3	11.7
BM-CIEH:L8-BO (THF, 120°C)	0.894	21.75	21.02	67.1	13.1

^aIntegrated from EQE spectra.

Reviewer #2:

In this manuscript entitled ‘Strongly H-aggregated small molecule donor for non-aromatic green solvent processed highly-efficient and stable all-small-molecule organic solar cells’ the authors achieve record power conversion efficiencies (15.0% in binary and ternary 16.1% in ternary) for green solvent processed all-small-molecule organic solar cells. By comparing three small molecule donors, the authors emphasize that strong H-aggregation behavior can maintain pre-aggregation state in THF, leading to good film-forming morphology and finally reflecting on high device performance. The effort from co-authors is appreciated in boosting the performance of the small molecule solar cells. However, there are severe novelty and urgency concerns raised when I looked at the manuscript. Some of the comments to improve the manuscript for further submissions can be found below:

1. The selection of THF as a green solvent should be re-considered, what is the motivation behind using THF as green solvent, what justifies that it is a novel green solvent? Capello Christian et al. reported a thorough assessment of green solvent (Green Chem. 2007, 9, 927-934. <https://doi.org/10.1039/B617536H>). THF is not recommendable from an environmental perspective: it causes high environmental impacts during petrochemical production; it has a complex petrochemical production route requiring several production steps. However, on line 107, the authors claim THF is cheap and environmental friendly, and do not depend on petroleum products without any citation.

Reply: Thanks for the reviewer’s comment. In our initial experiments, we employed toluene (commonly used as green solvent to process polymer solar cells) as processing solvent to fabricate BM-CIEH:BO-4Cl active layer, unfortunately, corresponding ASM-OSCs didn’t work well. Unlike blend solution of PD:SMA, the viscosity of SMD:SMA blend solution is smaller, which caused active layer thickness of AMS-OSCs to be lower at the optimal rotational speed if a high boiling point solvent such as toluene (or xylene) is used. Considering the boiling point (BP, 66°C) of THF is similar to that of CF (61.2°C)

and good solubility to BM-CIEH, BM-HEH and BO-4Cl, we tried to employ THF as processing solvent to fabricate BM-CIEH:BO-4Cl active layer. As a result, ASM-OSCs worked well and even performed better than CF-processed devices. We note that THF is not a novel green solvent but a general non-aromatic green solvent for processing OSCs (references: Adv. Mater. 2017, 29, 1604241; J. Mater. Chem. A 2019, 7, 23008; Adv. Mater. 2018, 30, 1704837; Adv. Funct. Mater. 2022, 32, 2107567, Adv. Funct. Mater. 2023, 2300778).

After carefully reading the reference recommended by the reviewer, we think it is more suitable to delete this claim like “THF is cheap and environmental friendly, and do not depend on petroleum products” in revised manuscript. THF may not be the best choice among many green solvents, but it is an alternative low boiling point green solvent with relatively excellent properties which are favorable for the film formation kinetics of ASM-OSCs (Adv. Funct. Mater. 2023, 2300778; J. Mater. Chem. A 2019, 7, 23008).

The corresponding revision in Manuscript (yellow highlight) is listed below:

However, sudden increase of molecular aggregation will significantly reduce the solubility of BM-Cl in green solvent where tetrahydrofuran (THF) is chosen considering its saturated vapor pressure that relates to volatility is similar to that of CF for a proper film-forming kinetics.

2. The authors claim in the abstract ‘green solvent processed ASM-OSCs are rarely reported and their power conversion efficiencies (PCEs) are difficult to exceed 10%.’ Recently, THF-processed binary ASM-OSCs reaching 14% PCE have been reported (Adv. Funct. Mater. 2023, 2300778. <https://doi.org/10.1002/adfm.202300778>). However, this paper is not cited in the introduction section or Figure 4d. There are further work on green solvents (not particularly on small molecules yet significant breakthroughs in the field) yet the manuscript does not even touch base.

Reply: We thank the reviewer for your kind reminder and comment. We are so sorry to miss this important work about green solvent processed ASM-OSCs. This mistake is committed because we did not update the comparison data in Fig. 4d in time before the manuscript was submitted. Thanks again for your kind reminder, and now we have cited this work in introduction part and Fig. 4d. We also simply summarized some important work progress on OSCs processed by green solvent.

The corresponding revision in Manuscript (yellow highlight) is listed below:

However, green solvent processed ASM-OSCs are rarely reported and its power conversion efficiencies (PCEs) are very difficult to improve.

Over the years, considerable efforts have been put into exploring green solvent processing for OSCs including material design, film forming kinetics, micromorphology optimization and efficiency improvement.^{38,39} Recent advances show

that green solvent processed OSCs enabled by PD:SMA, all-polymer and all-small molecule systems have achieved over 18%, 17% and 14% PCEs, respectively.⁴⁰⁻⁴²

Fig. 4 | Device structure and performance of ASM-OSCs. (a) Device structure. (b) *J-V* curves for best-performing green solvent processed devices under TA. (c) Corresponding EQE spectra. (d) Comparison of ASM-OSCs processed by green solvents. (e) Device reproducibility of BM-CIEH:BO-4Cl (THF) devices under as-cast, TA and SVA conditions. (f) Hole/electron mobilities for BM-Cl:BO-4Cl (CF), BM-CIEH:BO-4Cl (CF), BM-CIEH:BO-4Cl (THF) and BM-HEH:BO-4Cl (THF) blend films under TA condition.

Table S10 Photovoltaic performance of reported ASM-OSCs based on green solvent processing.

Active layer	Processing solvent	V_{oc} (V)	J_{sc} (mA cm^{-2})	FF (%)	PCE (%)	Year	Ref.
BTR:PC ₇₁ BM	Toluene	0.91	11.2	72.3	7.46	2017	4
N(Ph-2T-DCN-Et) ₃ :PC ₇₁ BM	benzaldehyde	0.96	8.27	46.75	3.71	2014	5
DPP-EZnP-O:PC ₆₁ BM	o-Xylene	0.75	15.73	50.0	5.85	2015	6
DRTT-R:F-2Cl	THF	1.00	16.82	62.6	10.45	2019	7
ZR1-C3:L8-BO	THF	0.897	23.24	67.41	14.05	2023	8

3. The energy levels of BM-Cl, BM-CIEH, and BM-HEH are missing. Regarding to the extremely low PCE of BM-HEH (<1%), it is important to show that such low PCE is not from the energy level mismatch by replacing the substitution group from -Cl to -H. Precise energy level determination is very important using photoelectron spectroscopy.

Reply: Thanks for the reviewer's good suggestion. The energy level mismatch between SMD and SMA is likely to result in inadequate exciton dissociation in D/A interface and thus low efficiency. In order to eliminate the inefficiency of ASM-OSCs caused by the energy level mismatch factor, we conducted the ultraviolet photoelectron spectrometer (UPS) test according to your suggestion. The precise HOMO/LUMO energy levels of BM-Cl (CF), BM-CIEH (CF), BM-CIEH (THF), BM-HEH (THF), BO-4Cl (CF) and BO-4Cl (THF) were determined to be -4.89/-3.06 eV, -5.05/-3.18 eV, -5.06/-3.20 eV, -5.02/-3.17 eV, -5.60/-4.16 eV and -5.57/-4.14 eV, respectively. It can be observed that the HOMO/LUMO energy levels of BO-4Cl are significantly lower than these of BM-Cl, BM-CIEH and BM-HEH, and there is no energy levels mismatch problem. Moreover, the PL spectra of corresponding blend films show that good charge transfer can be achieved between SMD and SMA. We have added the energy levels of studied materials into manuscript:

The corresponding revision in Manuscript (yellow highlight) is listed below:

Photoluminescence spectra (PL) of BM-Cl (CF), BM-CIEH (CF), BM-CIEH (THF), BM-HEH (THF), BO-4Cl (CF) and BO-4Cl (THF) neat films without/with TA or SVA were measured. As shown in Supplementary Fig. 12, TA and SVA treatments are able to improve molecular aggregation and/or ordering to induce PL emission enhancements (for SMDs) or red-shifts (for BO-4Cl). On the whole, TA effect is more obvious than SVA which is beneficial to achieve better film morphology. Bandgaps (E_g s) of investigated materials are determined by calculating the intersection of absorption and PL spectra (Supplementary Fig. 13), which are 1.831, 1.866, 1.857, 1.851, 1.439 and 1.431 eV, respectively. In addition, ultraviolet photoelectron spectroscopy (UPS) measurements were performed in order to accurately probe energy levels of SMDs and SMA (Supplementary Fig. 14). The Fermi levels (E_F) and valence bands (VB) of above films were fitted to be -4.34/0.55, -4.56/0.49, -4.50/0.56, -4.52/0.50, -4.62/0.98 and -4.45/1.12 eV, respectively, corresponding to highest occupied molecular orbital energy levels (E_{HOMO}) of -4.98, -5.05, -5.06, -5.02, -5.60, -5.57 eV, respectively. The lowest unoccupied molecular orbital energy levels (E_{LUMO}) were also obtained as -3.06, -3.18, -3.20, -3.17, -4.16 and -4.14 eV, respectively, according $E_{LUMO} = E_{HOMO} + E_g$. Sufficient energy levels offsets between E_{HOMOS} or E_{LUMOS} of SMDs and SMA can effectively ensure exciton transfer. Interestingly, the Fermi level of THF-processed films elevates relative to CF-processed one and VB widens, especially for BO-4Cl, which is likely to facilitate hole transfer and transport. PL spectra of BM-Cl:BO-4Cl (CF), BM-CIEH:BO-4Cl (CF), BM-CIEH:BO-4Cl (THF) and BM-HEH:BO-4Cl (THF) blend films exhibited significant quenching with efficiencies reach more than 90% after TA and SVA, indicating good charge transfer from SMA to SMDs (Supplementary Fig. 15).

Fig. S14. UPS tests for (a) BM-Cl (CF), (b) BM-CIEH (CF), (c) BM-CIEH (THF), (d) BM-HEH (THF), (e) BO-4Cl (CF) and (f) BO-4Cl (THF) neat films.

4. The link between the device FF and the hole/electron mobility ratio is confusing. From Figure 4f, CB-CIEH (CF) has an imbalance mobility ratio of over 20, which can be deduced as strong bi-molecular recombination, however, the FF of the device is still high (69.8%). Furthermore, we can see that by changing the donors and solvent, the most obvious mobility change is electrons rather than holes. The reasons that may cause such acceptor mobility change should be further elaborated. Additionally, the caption of Figure 4f should be more precise and the authors should discuss 4f in the main text as well.

Reply: Thanks for the reviewer's careful reading and good suggestion. We have carefully checked the electron and hole mobility data and determined that they are correct. Examples that ASM-OSCs shows relatively large electron/hole mobility ratio but also achieve relatively high FF can be seen, such as BTR:BO-4Cl that shows a

mobility ratio of 29.8 and achieves a FF of 67%; B1:BO-4Cl that shows a mobility ratio of 5.6 and achieves a FF of 75% (Sci. China Mater. 2020, 63, 1142-1150). To find out whether such a mobility ratio will result in severe bimolecular recombination, J_{sc} under different light intensities were measured shown in Fig. R2. It can be seen from the test result that ASM-OSCs based on BM-ClEH:BO-4Cl (CF) under TA treatment do not show serious bimolecular recombination despite its mobility ratio over 20. However, its bimolecular recombination is indeed a little more serious than other devices, so FF will be slightly lower.

Fig. R2. J_{sc} under different light intensities for ASM-OSCs based on BM-Cl:BO-4Cl (CF), BM-ClEH:BO-4Cl (CF) and BM-ClEH:BO-4Cl (THF) with TA or SVA treatments.

BM-Cl:BO-4Cl (CF) blend film shows best lamellar packing and π - π stacking among four TA blend films (Supplementary Table S14), so it exhibits highest electron mobility. With crystallization property of SMDs decreasing and solvent changing, lamellar packing and π - π stacking of BM-ClEH:BO-4Cl (CF), BM-ClEH:BO-4Cl (THF) and BM-HEH:BO-4Cl (THF) blend films will become weaker. The electron mobilities of blend films will be greatly affected and drop fast while change of hole mobilities of blend films are not very obvious. The increased hole mobility of BM-ClEH:BO-4Cl (THF) blend film may be due to favorable donor phase. Considering the coherence of manuscript, if Fig. 4f is discussed separately, the readability of the article will be reduced. I hope the reviewer can understand. Moreover, in morphology part, we also discussed the change of mobility.

The corresponding revision in Manuscript (yellow highlight) is listed below:

With crystallization property of SMDs decreasing and processing solvent changing, electron mobilities of blend films will be greatly affected (Fig. 4f) due to weaker lamellar packing and π - π stacking (Supplementary Table 14).

5. From Table 1, all as-cast devices have low PCE (<5%) no matter using CF or THF. All the high-performance devices are based on film thermal annealing treatment. Therefore, the link between the pre-aggregate state in solution and final device performance is not convincing and needs further discussion.

Reply: Thanks for the reviewer's comment. As-cast ASM-OSCs generally exhibits low PCE due that molecular packing and long-range ordering of SMD and SMA are destroyed during rapid film formation kinetics. High-performance ASM-OSCs almost all require TA or SVA post-treatments to assist donor and acceptor molecules move and rearrange. (*Nat. Commun.* 2019, 10, 5393; *Joule* 2019, 3, 3024; *Angew. Chem. Int. Ed.* 2020, 59, 2808; *Energy Environ. Sci.* 2020, 13, 1309). Morphology quality of TA- or SVA-treated films is closely related to that of as-cast films because TA or SVA induces range of molecular motion is very limited in a short time and solid state (such as BM-HEH:BO-4Cl, TA and SVA do not work). however, quality of as-cast films is greatly related to the pre-aggregation state of molecules in solution. So, the final device performance is related to pre-aggregation of molecule in solution, but not so directly. TA and SVA effects on PCE improvement is so obvious that lead us to easily ignore its role. Although all as-cast devices have low PCE (<5%) no matter using CF or THF, there is still a big difference in their efficiency: BM-Cl:BO-4Cl (CF)(5.1%)>BM-ClEH:BO-4Cl (THF)(3.0%)>BM-ClEH:BO-4Cl (CF)(2.4%)>BM-HEH:BO-4Cl (THF)(0.1%). This rule is consistent with crystallization properties of donor materials and pre-aggregation quality (temperature-dependent UV-vis absorption measurements in different solvents as stated in manuscript, which leads to CLs of lamellar and π - π stacking of as-cast films: BM-Cl:BO-4Cl (CF)>BM-ClEH:BO-4Cl (THF)>BM-ClEH:BO-4Cl (CF). The low PCE of BM-HEH:BO-4Cl (THF)-based AMS-OSCs is due to an inconspicuous phase separation.

6. In Figure 8d, the reason for testing the stability of the ternary device is lacking. A comparison between the green solvent processed device and the normal CF processed device will be useful.

Reply: Thanks for the reviewer's reminder and suggestion. Devices stability is one of key criteria to evaluate whether OSCs can be put into practical applications. Ternary ASM-OSCs based on BM-ClEH:B1:BO-4Cl (THF) shows green-solvent processing, high PCE, relatively good device repeatability (compared to PSCs), making it great potential for further application. Therefore, it is necessary to test its stability. The thermal stability of ASM-OSCs based on BM-ClEH:B1:BO-4Cl processing with THF and CF were measured (our MPP tracking system is down and has not been repaired, so, we use thermal stability instead of MPP tracking stability). Under continuous annealing at 80°C in a N₂ golvebox, THF-processed devices is able to maintain over 60% initial PCE after 200h, slightly better than that of CF-processing ones due to improved J_{sc} along with burning time.

The corresponding revision in Manuscript (yellow highlight) is listed below:

Device stability is another key criterion to evaluate the potential of OSCs for further applications. Stability test of optimal ternary device based on BM-ClEH:B1:BO-4Cl (weight ratio of 0.9:0.1:1.2) was carried out under thermal degradation and maximum power point (MPP) tracking (Supplementary Fig. 34 and Fig. 8d). Under a continuous

TA at 80°C in a N₂ glovebox without encapsulation, THF-processed devices are able to maintain over 60% initial PCE after 200 h, slightly better than that of CF-processed ones due to improved J_{sc} along with burning time. MPP tracking stability of ternary devices largely outperform thermal stability that can maintain over 80% initial PCE (T_{80}) after 200 h and 70% initial PCE (T_{70}) after 450 h, demonstrating great potential of BM-CIEH:B1:BO-4Cl device for commercial application.

Fig. S34. Thermal stability of CF and THF-processed ternary devices based on BM-CIEH:B1:BO-4Cl (0.9:0.1:1) under a continuous TA at 80°C in a N₂ glovebox.

Reviewer #3:

Gao et al report the “pre-aggregation” effects on the device performances fabricated by THF, and the ternary solar cells lead to an efficiency exceeding 16%. Relating the molecular structure with morphology is quite important. However, honestly speaking, the morphology for the ASM-OSCs is quite complicated. Because it is not only influenced by the solvent effect in solution, but also other many important factors such as Flory-Huggin parameter, crystalline, thermal effect, and solvent effect in film formation and evaporation. Hence, the whole paper doesn’t persuade me from the following two main aspects: a) What is the relationship between molecular structure with H-aggregates? Whether there are really H aggregates in your solution? b) Pre-aggregation for polymer is quite important, due to its limited diffusion speed during film formation. however, for small molecules, from my opinion, neglecting the miscibility and film formation process to talk about morphology is not reasonable. Therefore, I don’t recommend this paper to publish in Nature Communications, and my detail questions are as follows:

Reply: Thanks for the reviewer's comments. The relationship between H-aggregation and molecular structure is not very regular. For example, Ghosh and coworkers designed and synthesized a series of perylene bisimide (PBI) derivatives by changing peripheral alkyl chain on PBI. It was found that alkyl chain type (straight or bifurcation, bifurcation site and length) and substitution site will lead to PBI molecules to adopt H-aggregation, J-aggregation or a mixture of the two (H-/J-aggregation) (*Chem. Eur. J.* **2008**, *14*, 11343); Li et al. designed and synthesized a small molecule β CBF₂, it can grow two kinds of crystals, the green one adopts a total J-aggregation while the red one adopts a total H-aggregation (*Angew. Chem. Int. Ed.* **2021**, *60*, 18059). S. Kim et al. designed and synthesized four molecules named DCE4T, CE4T, DCB4T and CB4T, interestingly, the asymmetric molecules CE4T and CB4T show a total H-aggregation and symmetric ones DCE4T and DCB4T show H-/J-aggregation. (*Adv. Funct. Mater.* **2011**, *21*, 1616). Moreover, different additives can also cause molecule to form H-aggregation and/or J-aggregation (*J. Chem. Mater.* A DOI: 10.1039/D0TA11146E). So, the relationship between molecular structure with H-aggregation is hard to draw a conclusion. What tells us is that H-aggregation, J-aggregation or H-/J-aggregation need to be judged according to absorption behaviors of molecules. For the question "Whether there are really H aggregates in your solution?", we will answer it in question 1.

Miscibility between SMD and SMA, especially in BTR:Y6 system, have been widely studied. Miscibility problem will raise only when we observe large phase separation and/or domain size within active layer. Only in that time, we will consider the unfavorable morphology may be caused by the poor miscibility. For normal morphology (as we have observed in this work), miscibility difference is just used to assist in explaining whether the domains of donor and acceptor are evenly distributed and whether their sizes are appropriate for different control systems. In this work, we only observe very small difference in these aspects. Moreover, due to similar chemical structure of BM-CIEH and BM-HEH, the surface energy will be very close and their miscibility with BO-4Cl will be similar (this will be given in question 4). So, we think miscibility is not the key factor causing morphology difference between BM-CIEH:BO-4Cl and BM-HEH:BO-4Cl.

Film formation process of active layer is very fast and complicated, which makes it very difficult to study it clearly because it involves a lot of in-situ testing. We hope the reviewer can understand us that it is hard for us to explore all the questions clearly in one work.

1. In the absorption spectrum especially in solution, why it could use I(0-0) and I(0-1) to analyze the H and J aggregates? Which peak is attributed to internal charge transfer? In the film, how do you distinguish the differences between H/J aggregates and ICT absorption/aggregation absorption band? In fact, for small molecules, I am not convinced that it adopts H or J aggregates in the solution, although it definitely exists

solvation effect. The absorption spectrum variation with temperature in the diluted solution is maybe only because of rotation, and has nothing to do with H/J aggregates.

Reply: Thanks for the reviewer's comment. As stated by Frank C. Spano and Carlos Silva (*Annu. Rev. Phys. Chem.* 2014, 65, 477) "The classification of H- versus J-aggregation was a key development in unraveling the relationship between morphology and photophysical properties. The two aggregate types are depicted in Figure 1. In J-aggregates, neighboring chromophores are oriented in a head-to-tail fashion, resulting in a negative excitonic coupling and the placement of the optically allowed ($k = 0$) Frenkel exciton at the bottom of the exciton band. Conversely, in H-aggregates, nearest-neighbor chromophores are oriented in a more side-by-side manner, resulting in a positive excitonic coupling and the placement of the $k = 0$ exciton at the top of the exciton band. The energetic ordering of the excitons has profound effects on the photophysical response: In J- (H-) aggregates, the main $S_0 \rightarrow S_1$ absorption peak undergoes a red (blue) shift compared to the spectrum of an isolated monomer." "As outlined in Reference 35 (*Accounts Chem. Res.* 2010, 43, 429.), in J- (H-) aggregates, the ratio of the oscillator strengths of the first two vibronic peaks in the absorption spectrum, I^{0-0}_A/I^{0-1}_A , increases (decreases) with exciton bandwidth." When molecule behaves as J-aggregation, the oscillator strength of I^{0-0}_A is significantly stronger than that of I^{0-1}_A along with absorption spectra red-shifts (that is 0-0 exciton bandwidth become wider), so, the ratio of I^{0-0}_A/I^{0-1}_A increase. At this time, the strength of 0-0 peak is larger higher than that of 0-1, so, we can simply use 0-0 peak to judge the molecule as a J-aggregation. In ture, when molecule behaves as H-aggregation, the oscillator strength of I^{0-1}_A is significantly stronger than that of I^{0-0}_A along with absorption spectra blue-shifts (that is 0-1 exciton bandwidth become wider), so, the ratio of I^{0-0}_A/I^{0-1}_A decrease. At this time, the strength of 0-1 peak is larger higher than that of 0-0, so, we can simply use 0-0 or 0-1 peak to judge the molecule as a J or H-aggregation in solution or in film. We will added this into revised manuscript.

The corresponding revision in Manuscript (yellow highlight) is listed below:

H- and J-aggregations are two conventional aggregation modes of organic semiconductor materials. Due to different molecular stacking fashion, the S_0 - S_1 absorption transition will undergo a blue/red shift compared to that of an isolated monomer in H-/J-aggregates, which is used to classify H- versus J-aggregation by observing the ratio of 0-0 (vibrational transition) peak to 0-1 (vibrational transition) peak decreasing or increasing with bandwidth. When the 0-1 peak dominates, it indicates H-aggregation. Otherwise, it indicates J-aggregation.^{53,54}

As we know that the absorption bands of σ - σ^* and n - σ^* , n - π^* and π - π^* transitions are generally below 400 nm, and their absorption coefficient is much weaker than the ICT state. So, we usually classify strong absorption peaks with an absorption range of more than 400 nm as caused by ICT state's absorption. When there is no interaction between molecules, it is easy to observe complete ICT state absorption, but completely free molecules are difficult to exist. When there is an interaction between molecules (usually

in the form of dimers, trimers or multimers), the absorption of ICT states will appear in fine structures, such as 0-0, 0-1 and 0-2 peaks.

According to answer to the above two questions, we know that ICT state's absorption band is strong absorption peaks with an absorption range of more than 400 nm. H/J aggregates will cause ICT state's absorption to occur 0-1 and 0-0 absorption peaks.

Adding different proportions of poor solvent to the dilute solution of small molecules will generate more aggregates, which can be used to judge which aggregation form of small molecules adopt, J- or H- aggregation, according to the change of absorption spectra. Another judgment whether small molecules adopt a J-aggregation or a H-aggregation is to observe absorption changes of small molecule dilute solution with temperature. (Chem. Eur. J. 2008, 14, 11343; Angew. Chem. Int. Ed. 2011, 50, 3376). If J- and H-aggregates do not exist in small molecule's solution, all related studies will be wrong. Moreover, if there is no J- and H-aggregates in small molecule's solution, how to explain the enhancement or weakening, blue shift or red shifts of the 0-0 and 0-1 peaks in their absorption spectrum with proportions of poor solvent or different temperature. The absorption spectra of BM-CIEH and BM-HEH in solution show similar 0-0 and 0-1 peaks with these in films, moreover, the 0-0 peak of temperature-variable UV-vis absorption spectra of BM-HEH decrease and blue shift with temperature increased, while 0-1 peak of temperature-variable UV-vis absorption of BM-CIEH can remain almost unchanged due to strong H-aggregation of BM-CIEH that is able to tolerate solvent effect. These phenomena can well demonstrate that BM-CIEH in solution exists H-aggregates.

You can see from the temperature-variable absorption spectrum of BO-4Cl (Fig. S4) that the absorption spectrum of BO-4Cl in CF solution basically does not change as the temperature increases. While BO-4Cl in THF, with the increase of temperature, its 0-0 absorption peak decreased obviously and blue-shift. The single bond of BO-4Cl should have the same rotation in CF and THF solution as the temperature increases, the only explanation for why the absorption spectra are so different is the difference in the solvation effect of THF and CF on J-aggregates. It can be also seen from the temperature-variable NMR spectra of BM-CIEH and BM-HEH that the chemical shift changes very small with the increase of temperature, which is caused by the stretching vibration and rotation of hydrogen atoms which just requires low thermal energy to drive. If the single bonds within BM-CIEH and BM-HEH molecular skeleton is to be rotated, very high thermal energy is required, and the chemical shift of H atoms will be large. Also, if the changes of temperature-variable absorption spectra of BM-CIEH and BM-HEH is only due to the rotation of single bonds in the molecule, then how to explain the decrease of the 0-0 peak of BM-HEH and the invariance of the 0-1 peak of BM-CIEH at the same temperature. Therefore, the change of the temperature-variable absorption spectra of BM-CIEH (or BM-Cl) and BM-HEH is mainly caused by the H-aggregation or J-aggregation of molecules.

2. Before the part “Morphology, mobility and overall interaction for neat films” the author demonstrated that Cl-S non-covalent interaction within BM-Cl and BM-CLEH enhances molecular rigidity as a result to obtain strong H-aggregation property. My question is what is the relation-ship between molecular rigidity and H-aggregation property? Additionally, whose intermolecular force is larger, in your opinion, H or J?

Reply: Thanks for the reviewer’s comment. As we stated in manuscript “Considering that the interaction strength between molecules is proportional to the size of molecular interaction area, thus, H-aggregation with face-to-face parallel aggregation will have stronger intermolecular interactions than J-aggregation that adopts a dislocation parallel aggregation.” In *Angew. Chem. Int. Ed.* 2021, 60, 18059 paper, β CBF2 can grow two kinds of crystals, the green one adopts a total J-aggregation while the red one adopts a total H-aggregation. As stated in this work “It should be noted that the distorted molecular structure leads to loosely molecular packing in green crystals that possess a density of 1.209 g/cm³, while planar structure give rise to tight molecular packing in red crystals as evidenced by a larger density of 1.536 g/cm³”. This is a good example to show intermolecular force of H-aggregation is stronger than J-aggregation.

Based on our experiments on complete comparison with no Cl substituted BM-HE and previous knowledge that the optimal configuration of intermolecular aggregation follows the principle of lowest energy and the Cl-S non-covalent interaction can improve the planarity and rigidity of BM-CIEH (DFT calculation and temperature-variable NMR test), which can reduce the recombination energy between molecules and facilitate the formation of tight packing (H-aggregation), so, we can safely draw this conclusion “the author demonstrated that Cl-S non-covalent interaction within BM-Cl and BM-CLEH enhances molecular rigidity as a result to obtain strong H-aggregation property”, which is suitable for BTR type SMDs. It can be only said that the more rigid and planar the molecule is designed, the greater the probability of the molecule adopting a H-aggregation.

3. The H shifts with temperature in NMR is used to explain the molecular rigidity. What is your concentration for NMR (10 mg/ml)? What is your concentration for UV-vis (10⁻⁵ mg/ml) ? If there is a pre-aggregation in UV-vis absorption, and there would be strong pre-aggregation in your NMR test. It is puzzled me that the variation of H shifts with temperature is only related to molecular rotation and the variation of absorption spectrum with temperature is only related to pre-aggregation in the analysis.

Reply: Thanks for the reviewer’s comment. The concentration of samples for NMR test have been mentioned in SI and is 5 mg/mL. The concentration of samples for UV-vis test is about 10⁻⁵ mmol/mL. The pre-aggregation in NMR test will be much stronger than that in UV-vis absorption due to much higher sample concentration. But why the chemical shift changes of H atoms with temperature can be only attributed to the vibration and rotation of H atoms without considering the influence of molecular pre-aggregation is because the principle of nuclear magnetic resonance (NMR) determines

that pre-aggregation (or concentration) will not affect the chemical shift of H. NMR refers to the resonance transition phenomenon of atomic nuclei with magnetic moments excited by electromagnetic wave in a static magnetic field. When the hydrogen nucleus is in an external magnetic field, its external electrons will generate an additional magnetic field opposing the external magnetic field while rotating around the nucleus on a plane perpendicular to the external magnetic field. The additional magnetic field weakens the effect of the external magnetic field on the hydrogen nuclei, and this shielding effect that related to chemical environment of H atom is the source of the H chemical shift. When a hydrogen atom is heated and rotation, its chemical environment will change, resulting in a change in the chemical shift of H. The molecular aggregation has no effect on the state of H nucleus. So, we do not consider the influence of sample concentration on chemical shift of H atom when doing NMR test.

4. In BM-HEH:BO-4Cl, its poor efficiency may be due to its unsatisfactory hetero-intermolecular interaction, and it may not be related to the "pre-aggregation" you mentioned. Both the halogen substituent and the alkyl chain modification have a great influence on the surface energy.

Reply: Thanks for the reviewer's comment. To answer this question, we conducted contact angle tests (Fig. R3). The contact angles with water and EG as the medium and corresponding surface tension obtained by Wu' mode are listed in Table R1. From this table, we can see that halogen substitution and alkyl chain modification will affect the surface energy of molecule, but this effect is not significant. According to Flory-Huggin parameter χ calculation formula: $\chi \propto (\sqrt{\gamma_A} - \sqrt{\gamma_B})^2$ where γ_A and γ_B are the surface energy of two materials, we can know that when the surface energies of two donor materials are similar, their χ is similar when blended with the same acceptor materials. The χ parameter of BM-ClEH:BO-4Cl (THF) is 1.38 times that of BM-HEH:BO-4Cl (THF). I think such a small gap will not cause too much difference in the hetero-intermolecular interaction between BM-ClEH:BO-4Cl and BM-HEH:BO-4Cl. Our tests (AFM and TEM) show that the reason why BM-HEH:BO-4Cl (THF) does not work is because there is no obvious phase separation between BM-HEH and BO-4Cl, resulting in no free charge generation. And the reason for no phase separation is that the pre-aggregation state of BM-HEH is not as good as BM-ClEH in THF solution, which results in poor film quality.

Fig. R3. Contact angle test with water and EG as the medium for BM-CI (CF), BM-CIEH (CF), BM-CIEH (THF), BM-HEH (THF), BO-4Cl (CF) and BO-4Cl (THF) films.

Table R1 Contact angles and corresponding surface energy obtained by Wu' mode for BM-CI (CF), BM-CIEH (CF), BM-CIEH (THF), BM-HEH (THF), BO-4Cl (CF) and BO-4Cl (THF) films.

Samples	CA for water (°)	CA for EG (°)	Surface tension ^{a)} (mN m ⁻¹)
BM-CI (CF)	104.8	77.3	23.2
BM-CIEH (CF)	104.3	77.4	22.7
BM-CIEH (THF)	102.5	76.4	22.3
BM-HEH (THF)	104.5	76.7	23.6
BO-4Cl (CF)	100.2	72.2	24.5
BO-4Cl (THF)	102.1	73.5	24.6

5. For the BP (boiling point) for THF is higher than CF, its film is always more condensed than that spin-coated by CF, which shouldn't be contributed this only to the solvent-effect in solution. Besides, as for BO-4Cl (THF) film undergoes largest thermal motion deduced from MDT, however, its electron mobility is only twice times higher,

which is much inferior compared to others (one order magnitude improvement after TA, Figure 3). So, in your opinion, what is the key point to induce thermal packing?

Reply: Thanks for the reviewer's comment. According to the following question, we guess "more condensed" you mentioned should mean such a film with a smaller DM_T value and a slower change rate of DM_T with temperature. So, your question can be understood as THF spin-coated film has greater overall interaction between molecules than CF spin-coated one not only because of solvent-effect (understood as pre-aggregation of molecules) in solution but also because of higher BP of THF. As we mentioned in the manuscript "Changes in crystallinity and pre-aggregation state of involved materials and differences of processing solvents in vapor pressure will have an impact on thermodynamic and kinetic factors of transition from solution to solid film, thus resulting in different morphology of films". So, "More condensed" film may be caused by many aspects. For example, BO-4Cl (THF) film is "less condensed" than BO-4Cl (CF) due to poor pre-aggregation of BO-4Cl in THF; BM-Cl (CF) film is "more condensed" than BM-ClEH (CF) film due to strong crystallinity of BM-Cl because BM-Cl and BM-ClEH show similar pre-aggregation state in CF; BM-ClEH (THF) film is "more condensed" than BM-ClEH (CF) due to the processing solvent difference because BM-ClEH has similar pre-aggregation in CF and THF.

Thermal annealing can lead to thermal movement of molecules to improve molecular packing and ordering, thereby improving carriers mobility. But the extent to which this thermally induced movement is able to improve molecular packing and ordering varies from molecule to molecule (SMA and SMD). As for the extent to which electron mobility can be increased, different materials may be different. The electron mobility of BO-4Cl (THF) as-cast film is measured to be $1.99 \times 10^{-3} \text{ cm}^2\text{V}^{-1}\text{s}^{-1}$. If its electron mobility can be increased by more than an order of magnitude after TA, then its electron mobility will be above $2.0 \times 10^{-2} \text{ cm}^2\text{V}^{-1}\text{s}^{-1}$. I have never seen such a high electron mobility of a non-fullerene acceptor measured by SCLC method even for polymer donors.

The key to inducing thermal packing lies in the as-cast film morphology that related to pre-aggregation in solution. For as-cast films with poor initial morphology, suitable TA temperature and time can help molecule to form better molecular packing, such as BO-4Cl (THF) film. For as-cast films with good initial morphology, TA will result in poor molecular packing, which can be found in many high-performance as-cast OSCs.

6. The TAS part is not well organized, although the absorption of intermediate states covering 700-800 nm seems interesting, however, the explanation is not convincing. From time-scale, it emerges even faster than the GSB signal of donor, how does this happen? Further, there are many small errors in this part. Such as "GBS", "stark effect then EA features" "overlap with the EA on the said range" and so on. In Figure S23, what are the molecules of DG7, DG8? Figure S22, what is the wavelength of singlet exciton? In stability measurements, what is the condition? N2 atmosphere or exposure to the air after encapsulation?

Reply: Thanks for the reviewer's careful reading and insightful comment. The broad negative features spanning from 700-800 nm are well known to be arising from polarons photo-induced absorption (PIA) wherein the donor electroabsorption (EA) partly overlaps. The separation of excitons will form positive and negative charges imparting a weak localized electric field thereby disrupting the electron density distribution and causing stark effects. This effect will give rise to the absorption derivative-like feature (known as EA) in the transient absorption wherein the donor EA leads to negative features around 720 nm. In fact, these concepts are already widely known and reported, for instance in this study (*Energy Environ. Sci.*, 2022, 15, 1545-1555) where they are clearly marked in the supplementary details. The emergence of donor EA can be an ultrafast time scale (i.e., can be faster than our instrument response) as certain neat polymer films have previously been observed to display ultrafast intramolecular exciton splitting generating holes and electrons in the donor moieties (*J. Am. Chem. Soc.* 2012, 134, 9, 4142-4152). Hence, the donor EA features can practically appear at a very early time scale just like the donor singlets. Indeed, there are an increasing number of reports observing homojunction organic materials capable of such intra-moiety charge generation cases (*Nature Communications* 2020, 11, 4617; *Adv. Opt. Mater.* 2017, 5, 1700024). A very recent paper also mentioned this EA using donor:PCBM blends and illustrated that charge generation causes these deep negative features (*Energy Environ. Sci.*, 2023, 16, 3416-3429). Consequently, as transient absorption signals are proportional to charge population density, we then believe that it is convincing enough to speculate that deeper EA upon TA and SVA imparts more efficient donor intra-moiety charge generation forming more free charges.

We have corrected the typo "GBS" into "GSB" in revised manuscript. The corresponding probe ranges of the singlet excitons for Fig. S30 (650-690 nm) were mentioned in the manuscript ("The maximum GSB for SMDs and acceptor are found to be located at 620-680 nm and 650-690 nm, respectively, thereby used as the corresponding probe ranges in the following investigations"), but for better clarity, we mentioned them again in the supplementary figure captions.

We are so sorry to make this mistake. DG7 is BM-CIEH and DG8 is BM-HEH. DG7 and DG8 are the names we gave to the SMDs after they were synthesized, while BM-CIEH and BM-HEH were named when we compiled the manuscript. We have corrected them and have also checked the whole manuscript. The thermal and MPP tracking stability were both measured in a N₂ glovebox without encapsulation, which have been added into revised manuscript.

Fig. 31. Spectral cuts of TAS measurements after different time for blend films under different conditions.

REVIEWER COMMENTS

Reviewer #1 (Remarks to the Author):

This is a fascinating topic, and the results are impressive. Publish as is, as all revisions have been carefully included.

Reviewer #2 (Remarks to the Author):

The authors commented on several aspects of the reviewers comments and suggestions and added experiments. However, the whole concept of their claims are not fully justifiable even after these experiments I do feel. A lot of the comments are I feel word puzzles and not to the point-by-point.

Several claims are looking to satisfy the reviewer however does not improve and strengthen the claims they offer.

I understand from other reviewer comments that there are concerns on the pre-aggregation state in solution which is quite critical as authors claim. However there is no direct evidence on that beyond absorption with complex yet analytical techniques such as cryo-EM etc.

The concept of green solvent as indicated is withdrawn to certain extent and authors accept that their performances are not par with solvents such as xylene. When we consider scale-up or commercialization of such technologies, there is a clear no-go using solvents such as toluene or THF from industry relevance. And I find it quite disrupting in the research community to keep publishing such stories claiming we use 'green' solvents. Just to state the fact THF is a non-aromatic. not green.

Further looking into title of 'stable' is another fish story used as a buzz word in the title, but when you look into figure 8, T80 is 200h. What is the term 'stability' mean to authors? Would they call a 40% performance decrease in a cell over 500h a 'stable' device if they review a paper for a journal like Nature Comm?

Not to mention there are 8 main figures, which is totally overstated to tell a story and not common, to push a certain perspective with several proof.

Combining all these factors make me question what is the novelty and importance of this story to be published in such journal?

I did really try to empathize on the concept of this story, however I do stand by my opinion on that the paper does not meet the merits for Nature communications overall.

Reviewer #3 (Remarks to the Author):

See Attachment

Most of the questions have been stated, and authors tried their best to do lots of characterizations to elaborate the pre-aggregation, TA and SVA effects on the morphology and ultimately the device performances. The paper still needs following revisions before accept:

1) The title is "Strongly H-aggregated small molecule donor for non-aromatic green solvent processed highly efficient and stable all-small molecule organic solar cells". The "stable" needs further consideration because the T_{80} of thermal stability for ternary solar cells is less than 100 hours, and the T_{80} for MPP track without light-soaking or thermal annealing is also just so so. Furthermore, the "Strongly H-aggregate" is judged by the I_{0-1} , but usually the I_{0-0} represents the most red-shifted peak, and the I_{0-1} is in the following. However, as shown in Figures 1d and 1e, for the BM-HEH absorption spectrum, in addition to the (0-0) peak signed by the author, there is an obvious red-shifted peak in the film, which represents the (0-0) peak in many other literatures. Additionally, as shown in Figure S9, the spectra of BM-HEH and BM-CIEH do not show much difference, especially between 80-120°C. So, what are your criteria for judging (0-0) and (0-1) in the absorption spectrum? Since the title is "strongly H-aggregate", the judge of H-aggregate must be persuadable.

2) In many literatures, the high absorbance of (0-0) attributed to J-aggregation in the film is applied to confirm the strong or ordered molecular packing (ACS Appl. Mater. Interfaces 2021, 13, 9, 11108; ACS Appl. Mater. Interfaces 2019, 11, 47, 44528), and also, they exhibit good device performances. This is conflict with your opinion, what is your explanation about this? So, in your work, essentially speaking, is the H or J aggregation dominated the difference in morphology/device performance or the packing ability dominated?

3) The authors try to relate the pre-aggregation with TA/SVA effects through their effect on the T_g and slope of DM_T in the initial film. Generally, a small T_g accompanies with a more sensitive packing to thermal annealing. However, as author demonstrate that the BO-4Cl in THF is more H-pre-aggregation, but it exhibits higher T_g with a much larger slope. What is your explanation for this?

4) Small errors. Such as:

a) In Figure 2, the temperature range is 0-40°C, which is different from the measurements' part (Range of test temperature for NMR (Bruker AVANCE 400 MHz) spectra is from 20 °C to 56 °C) ;

b) It is worth noting that BO-4Cl (THF) film undergoes largest thermal motion with MD_T value of up to 7 and K_2 of 0.124 (Fig. 3j), what is MDT represents for?

c) In the supporting information, some use Fig. S and some use Fig. (such as Fig. 29 and so on)

Reviewers' Comments to Authors:

Reviewer #2:

The authors commented on several aspects of the reviewers comments and suggestions and added experiments. However, the whole concept of their claims are not fully justifiable even after these experiments I do feel. A lot of the comments are I feel word puzzles and not to the point-by-point. Several claims are looking to satisfy the reviewer however does not improve and strengthen the claims they offer. I understand from other reviewer comments that there are concerns on the pre-aggregation state in solution which is quite critical as authors claim. However there is no direct evidence on that beyond absorption with complex yet analytical techniques such as cryo-EM etc. The concept of green solvent as indicated is withdrawn to certain extent and authors accept that their performances are not par with solvents such as xylene. When we consider scale-up or commercialization of such technologies, there is a clear no-go using solvents such as toluene or THF from industry relevance. And I find it quite disrupting in the research community to keep publishing such stories claiming we use 'green' solvents. Just to state the fact THF is a non-aromatic. not green. Further looking into title of 'stable' is another fish story used as a buzz word in the title, but when you look into figure 8, T80 is 200h. What is the term 'stability' mean to authors? Would they call a 40% performance decrease in a cell over 500h a 'stable' device if they review a paper for a journal like Nature Comm? Not to mention there are 8 main figures, which is totally overstated to tell a story and not common, to push a certain perspective with several proof. Combining all these factors make me question what is the novelty and importance of this story to be published in such journal? I did really try to empathize on the concept of this story, however I do stand by my opinion on that the paper does not meet the merits for Nature communications overall.

Reply: We thank the reviewer for carefully reviewing our previous response and once again giving valuable comments.

It took a long time for us to further figure out the difference of pre-aggregation state of two SMDs in solution by beyond temperature-dependent UV-vis absorption

spectra. We first tried to make appointment for cryo-EM testing, but unfortunately failed even after a long time waiting, as cryo-EM equipment is very expensive and resources are very limited. Subsequently, we decided to use grazing incident small-angle X-ray scattering (GISAXS) measurement to explore the pre-aggregation behavior of two SMDs in solution THF, with the help from Professor Xinhui Lu of the Chinese University of Hong Kong. The experimental results are as follows:

Fig. R1. 2D-GISAXS images for (a) capillary and THF (background); (b) BM-HEH in THF solution (10 mg/mL); (c) BM-CIEH in THF solution (10 mg/mL). (d) corresponding cut-line profiles.

For thin film samples, the film can be peeled off from substrate and then tested by GISAXS. All the signals obtained come from sample itself. But for solution samples, both the container and solvent absorb and scatter X-ray, so the scattering signal of sample itself will be weakened, and it will take a long time of X-ray irradiation to obtain a strong enough scattering signal (48 hours for every sample in this test). From **Fig. R1d**, it can be observed that the scattering signal of BM-CIEH is slightly stronger than that of BM-HEH, but after deducting the scattering signal of background (capillary and THF), their intensities are very weak and cannot meet the fitting requirements, so there is no way to obtain the corresponding pre-aggregation scale of BM-CIEM and BM-

HEH in THF solution. In addition, we consider that the Cl atoms in BM-CIEH have a very strong absorption effect on X-ray, which will cause the scattering signal of BM-CIEH to be weak. With this in mind, we tried to find a GISAXS instrument with a stronger X-ray light source to retest. Subsequently, we contacted the Suzhou Institute of Nano-Tech and Nano-Bionics (SINANO) of the Chinese Academy of Sciences to conduct GISAXS testing based on its nano vacuum interconnection experimental station, where a stronger X-ray light source can be provided. Here are the test results.

Fig. R2. 2D-GISAXS images for (a) capillary and THF (background); (b) BM-HEH in THF solution (10 mg/mL); (c) BM-CIEH in THF solution (10 mg/mL). (d) corresponding cut-line profiles.

It can be seen from **Fig. R2d** that when the X-ray light source intensity increases, the difference in scattering signal intensity between BM-HEH and BM-CIEH becomes more obvious, but the background scattering signal is still strong. Unfortunately, after subtracting the background signal, the scattering signal intensity of BM-HEH and BM-CIEH itself cannot be used for fitting the pre-aggregation scale.

Due to time constraints in responding to reviewer's comments, we did not conduct further experiments.

For the other comments and suggestions, we have made the following changes to our manuscript, which are highlighted in yellow in the revised manuscript:

1. We have changed the term “THF as a green solvent” in the full text to a “non-halogen solvent” (different from chlorobenzene (aromatic) and chloroform (non-aromatic) halogen solvents).
2. We have removed the word “stable” from title and main text, and stability data is provided for reference for subsequent researches.
3. We have moved Fig. 8 to Supporting Information.

Overall, this work has enabled the efficiency of ASM-OSCs processed with non-halogen solvent to exceed 16%, which is the highest efficiency currently achieved for ASM-OSCs processed with non-halogen solvent. It also illustrates the importance of strong aggregation property of SMD for achieving efficient non-halogen solvent processing of ASM-OSCs. We enhance the rigidity of SMD through intramolecular Cl-S non-covalent interactions, resulting in strong aggregation of SMD with better solubility in THF. By comparing three materials, two processing solvents and three treatments, we tried to establish connections from material design, molecular aggregation characteristics, pre-aggregation state, film morphology to device performance to clarify the importance of strong aggregation property on achieving efficient non-halogen solvent processing of ASM-OSCs through various characterization methods, which provides guidance for subsequent work. We sincerely hope that the reviewer can re-review this work and our efforts in improving this work.

Reviewer #3:

Most of the questions have been stated, and authors tried their best to do lots of characterizations to elaborate the pre-aggregation, TA and SVA effects on the morphology and ultimately the device performances. The paper still needs following revisions before accept:

- 1) The title is "Strongly *H*-aggregated small molecule donor for non-aromatic green solvent processed highly efficient and stable all-small molecule organic solar cells". The "stable" needs further consideration because the T_{80} of thermal stability for ternary

solar cells is less than 100 hours, and the T_{80} for MPP track without light soaking or thermal annealing is also just so so. Furthermore, the "Strongly H-aggregate" is judged by the I0-1, but usually the I0-0 represents the most red-shifted peak, and the I0-1 is in the following. However, as shown in Figures 1d and 1e, for the BM-HEH absorption spectrum, in addition to the (0-0) peak signed by the author, there is an obvious red-shifted peak in the film, which represents the (0-0) peak in many other literatures. Additionally, as shown in Figure S9, the spectra of BM-HEH and BM-CIEH do not show much difference, especially between 80-120°C. So, what are your criteria for judging (0-0) and (0-1) in the absorption spectrum? Since the title is "strongly H-aggregate", the judge of *H*-aggregate must be persuadable.

2) In many literatures, the high absorbance of (0-0) attributed to *J*-aggregation in the film is applied to confirm the strong or ordered molecular packing (ACS Appl. Mater. Interfaces 2021, 13, 9, 11108; ACS Appl. Mater. Interfaces 2019, 11, 47, 44528), and also, they exhibit good device performances. This is conflict with your opinion, what is your explanation about this? So, in your work, essentially speaking, is the H or J aggregation dominated the difference in morphology/device performance or the packing ability dominated?

3) The authors try to relate the pre-aggregation with TA/SVA effects through their effect on the T_g and slope of DMT in the initial film. Generally, a small T_g accompanies with a more sensitive packing to thermal annealing. However, as author demonstrate that the BO-4Cl in THF is more H-pre-aggregation, but it exhibits higher T_g with a much larger slope. What is your explanation for this?

4) Small errors. Such as: a) In Figure 2, the temperature range is 0-40°C, which is different from the measurements' part (Range of test temperature for NMR (Bruker AVANCE 400 MHz) spectra is from 20 °C to 56 °C) ; b) It is worth noting that BO-4Cl (THF) film undergoes largest thermal motion with MDT value of up to 7 and K_2 of 0.124 (Fig. 3j), what is MDT represents for? c) In the supporting information, some use Fig. S and some use Fig. (such as Fig.29 and so on)

Reply: We thank the reviewer for carefully reviewing our previous responses, and we are also very grateful to the reviewer for giving us a valuable opportunity to revise.

We have removed the word "stable" from the title and main text. We tested the device stability (thermal and MPP stability) of ASM-OSCs only as reference data for subsequent related research.

The principle for judging *0-0* and *0-1* peaks is consistent with what you described. The most red-shifted absorption peak is *0-0* (vibrational transition) peak, and the first blue-shifted absorption peak following is *0-1* (vibrational transition) peak. The absorption peak of BM-HEH film is very broad, which may be caused by large overlap of *0-0* and *0-1* peaks, making it difficult to observe clearly. We performed differential processing on the absorption spectra of BM-HEH films (CF/THF) and found that this broad peak is indeed composed of two peaks (**Fig. R3**). Upon annealing, the *0-0* and *0-1* absorption peaks of BM-HEH film (CF/THF) become more obvious due to the enhanced packing between molecules. Thanks to the reviewer's valuable comments, we have recalibrated the positions of *0-0* and *0-1* peaks in **Fig. 1c** and changed the aggregation mode of BM-HEH in the film from *J*-aggregation to weak aggregation, because only after annealing the BM-HEH film absorption spectra can be consistent with that of BM-CIEH without annealing. In addition, DSC tests also show that the intermolecular interaction of BM-HEH is weaker than that of BM-CIEH.

Fig. R3. Absorption spectra of BM-HEH and BM-CIEH films (CF/THF) and corresponding first-order derivatives. (It can be observed from **Fig. R3** that the absorption peak can be approximated as a parabola, and its first-order derivative is an approximate straight line (500-570 nm). However, in the range from 570 to 620 nm, the first-order derivative curve turns, which proves that a new peak appears)

For BP-Cl, BM-Cl (BP-Cl and BM-Cl are two SMDs we reported previously, *Angew. Chem. Int. Ed.* **2022**, *61*, e202205168) and BM-CIEH, the order of their *H*-aggregation strength is probably BP-Cl > BM-Cl > BM-CIEH, their photovoltaic

performance is BM-Cl > BM-CIEH > BP-Cl. The poor photovoltaic performance of BP-Cl is caused by excessive molecular packing. The packing ability of BM-CIEH is weaker than that of BM-Cl, and the photovoltaic performance decreases.

For the reviewer's statements how/whether *H-/J*-aggregated SMDs can achieve high photovoltaic performance, our opinion is that it is necessary to appropriately increase molecular packing and stacking ordering to improve the morphology of blend films. The main factor that determines the morphology and photovoltaic performance of ASM-OSCs is the packing ability of SMD. Therefore, we changed the "strongly *H*-aggregated small molecule donor" to the "strongly aggregated small molecule donor".

The molecular packing of BO-4Cl in THF solution is more susceptible to be damaged as shown by the temperature-dependent absorption spectrum of BO-4Cl in THF solution compared to in CF solution (**Fig. S4**). During the film formation process, the residual THF will volatilize as the temperature increases, causing the stacking of BO-4Cl to become denser. This can be observed by comparing GIWAXS measurements of annealed BO-4Cl film (THF) and annealed BO-4Cl film (CF) (**Table S4**). Compared with CF, THF may play a similar role of additive to the BO-4Cl film, which helps to enhance stacking between SMD and increase the T_g of film.

The temperature ranges for temperature-dependent ^1H NMR measurements of BM-Cl, BM-CIEH and BM-HEH is from 20 °C to 56 °C. But to better observe the difference, in **Fig 2a-i**, we plotted with "temperature change" as the abscissa and "chemical shift change" as the ordinate. So, in **Fig 2a-i**, the range of the abscissa is 0-36 °C (taking 20 °C as the starting point, observing chemical shift change every 4 °C relative to the chemical shift at 20 °C). Deviation metric (DM_T) (this have been defined in the manuscript) is the sum of squared deviation in the absorbance between as-cast and annealed films:

$$DM_T = \sum_{\lambda_{min}}^{\lambda_{max}} [A_{RT}(\lambda) - A_T(\lambda)]^2$$

where λ_{min} and λ_{max} are the lower and upper bounds of optical sweep, $A_{RT}(\lambda)$ and $A_T(\lambda)$ are normalized absorption intensities of as-cast and annealed films, respectively. DM_T can be simply understood as the overall change of film absorption spectrum within a

certain temperature ranges. We have carefully checked the main text and SI and corrected the errors you mentioned and errors we found.

Corresponding changes have been highlighted in yellow in the revised manuscript.

REVIEWERS' COMMENTS

Reviewer #2 (Remarks to the Author):

The reviewers greatly appreciated the authors removed their exaggerated claims from the manuscript. In all honestly, with all claims removed this paper unfortunately does not meet most of the criteria for Nature Communications. It is also not the highest efficiency non-halogen OSC either. Please remove such claims unless you are 100% sure about these, which are also not much meaningful as we are all pushing the limits of this field.

I would accept this paper as there are a lot of efforts and still scientifically interesting and important.

I just want to clarify here as well that the field got very toxic and unfriendly in photovoltaics. The fact that a reviewer does not agree with the conclusions does not mean the work is not original and novel. There should be contradicting work so that others can take it onwards and make things better. In this regard, I accept this paper can be published in Nature Comm.

However there are certain things authors should make sure they mention in their paper:

- The importance of pre-aggregation and the fact that they explained with UV and GIWAX still does not mean these are direct methods and using cryo-EM or similar techniques would be very interesting for such direct observation.
- They should remove any claim saying this is the highest non-halogen efficiency.
- They also should refer to emerging-pv.org and cite those reports on the progress of OPV in the stability section mentioning the requirements for stability.

Reviewer #3 (Remarks to the Author):

All of questions I am concerned are reasonably demonstrated, I have no further questions. I recommend its publication.

Reviewers' Comments to Authors:

Reviewer #2:

The reviewers greatly appreciated the authors removed their exaggerated claims from the manuscript. In all honestly, with all claims removed this paper unfortunately does not meet most of the criteria for Nature Communications. It is also not the highest efficiency non-halogen OSC either. Please remove such claims unless you are 100% sure about these, which are also not much meaningful as we are all pushing the limits of this field.

I would accept this paper as there are a lot of efforts and still scientifically interesting and important.

I just want to clarify here as well that the field got very toxic and unfriendly in photovoltaics. The fact that a reviewer does not agree with the conclusions does not mean the work is not original and novel. There should be contradicting work so that others can take it onwards and make things better. In this regard, I accept this paper can be published in Nature Comm.

However there are certain things authors should make sure they mention in their paper:

-The importance of pre-aggregation and the fact that they explained with UV and GIWAX still does not mean these are direct methods and using cryo-EM or similar techniques would be very interesting for such direct observation.

-They should remove any claim saying this is the highest non-halogen efficiency.

-They also should refer to emerging-pv.org and cite those reports on the progress of OPV in the stability section mentioning the requirements for stability.

Reply: We are very grateful for the reviewer's pertinent comments and the reviewer's kindness.

We noted that UV-vis and GIWAXS measurements are common but not direct methods to explore the pre-aggregation state of materials in processing solvent. In further work, we plan to use cryo-EM instead of UV-vis and GIWAXS measurements to explore the pre-aggregation state of materials in processing solvent, which will help

us better and more directly study pre-aggregation state and its importance for the improvement of photovoltaic performance.

We have carefully checked the manuscript and removed/changed the claim "highest non-halogen efficiency" to "high efficiency" or "one of the highest efficiency" for non-halogen solvent processing of all-small-molecule OSCs.

We have referred to emerging-pv.org and cited those reports on the progress of OPV in the stability section mentioning the requirements for stability as following:

61. Almora, O. et al. Device Performance of Emerging Photovoltaic Materials (Version 1). *Adv. Energy Mater.* **11**, 2002774(2021).

62. Almora, O. et al. Device Performance of Emerging Photovoltaic Materials (Version 2). *Adv. Energy Mater.* **11**, 2102526(2021).

63. Almora, O. et al. Device Performance of Emerging Photovoltaic Materials (Version 3). *Adv. Energy Mater.* **13**, 2203313(2023).

64. Almora, O. et al. Device Performance of Emerging Photovoltaic Materials (Version 4). *Adv. Energy Mater.* **14**, 2303173(2024).